# LAGRANGIAN METHOD FOR EPISODIC LEARNING

## ABSTRACT

This paper considers the problem of learning optimal value functions for finite-time decision tasks via saddle-point optimization of a nonlinear Lagrangian function that is derived from the $Q$-form *Bellman optimality equation*. Despite a long history of research on this topic in the literature, previous works on this general approach have been focusing on a linear special case known as the *linear programming approach to RL/MDP*. Our paper brings new perspectives to this general approach in the following aspects: 1) Inspired by the usually-used linear $V$-form Lagrangian, we proposed a *nonlinear Q-form* Lagrangian function and proved that it enjoys strong duality property in spite of its nonlinearity. The Lagrangian duality property immediately leads to a new imitation learning algorithm, which we applied to Machine Translation and obtained favorable performance on standard MT benchmark. 2) We pointed out a fundamental limit in existing works which exclusively seeks to find the *minimax-type* saddle points of the Lagrangian function. We however proved that another class of saddle points, the *maximin-type* ones, turn out to have better optimality property. 3) In contrast to most previous works, our theory and algorithm are oriented to the *undiscounted* episode-wise reward, which is practically more relevant than the usually considered discounted-MDP setting, thus have filled a gap between theory and practice on the topic.

## 1 INTRODUCTION

Episodic learning Terry (2017) is a general learning paradigm in which the agent learns based on data collected from a sequence of episodes of environmental interactions. Each episode consists of a finite number of decision steps, and the goal is to maximize an episode-wise performance. With one-step episodes, episodic learning subsumes supervised learning and bandit problems as special cases; for sequential decision making problems, an episode may contain multiple (and possibly varying number of) steps, in which case episode learning encompasses most reinforcement learning and imitation learning problems encountered in current AI practice.

A pervasive idea of episodic learning, often called the *value-based approach* Sutton and Barto (2018), seeks to find a *value function* such that greedily selecting the actions with the best values (as prescribed by this function) gives good performance. For example, in most classification tasks, most deep neural networks are learned (and used) to represent such a value function. Finding good value functions can be highly challenging, especially for tasks with multi-step episodes. In this case, the classic *Bellman optimality equation* Bertsekas (2019) gives a sufficient condition of *optimal* value function (i.e. value functions with which greedy decisions have optimal performance).

In this paper, we study a *Lagrangian method* which learns optimal value functions by characterizing them as saddle points of the *Lagrangian function* of a variational (re-)formulation of the Bellman equation. The method reveals an elegant thoery on *policy-value duality* in machine learning. A restricted form of this high-level idea has been known as the Linear Programming (LP) approach in the literature of dynamic programming Puterman (1994); Si et al. (2004); Chen and Wang (2016); Wang (2017), which has also been introduced to reinforcement learning in recent years Dai et al. (2018); Lee and He (2018); Nachum et al. (2019); Yang et al. (2020); Nachum and Dai (2020). This paper provides a new perspective to the foundation of this elegant idea through the following contributions:

Firstly, previous works are mostly limited to the discounted-reward setting, which bypass some technical difficulties Nachum and Dai (2020), but deviate from the practical settings of many (if not

most) real-world tasks. This paper fills this well-known gap between theory and practice by establishing a value-based theory (and algorithm) that targets at directly learning optimal value functions and policies for the *undiscounted* episode-wise reward (see (1) in Section 2).

Secondly, previous works focused on *linear* formulations, which enjoys the strong duality property and generic LP techniques, but are consequently limited to treatments for the state-value functions (i.e. V-functions) in policy optimization setting. However, the optimal state-value function cannot be directly used to derive optimal policy in model-free learning. In contrast, we proved a *minimax theorem* for the *nonlinear* Lagrangian functions associated with the action-value functions (i.e. Q-functions). The theorem opens a door to nonlinear variational treatment to the Bellman equation, which now enjoys the strong duality property too (see Section 3).

In particular, we developed a simple imitation learning algorithm based on the Lagrangian duality thus established, and applied the algorithm to Machine Translation (MT) as case study. Transformer models trained by our algorithm achieves beam-search-level performance Vaswani et al. (2017) with only greedy decoding, and leads to $1.4$-BLEU improvement when also equipping with beam search (see Section 4).

Last but not the least, previous works have exclusively focused on the *primal-form optimization* formulation of the Bellman equation, thus are limited to solving the minimax-type saddle points of the corresponding Lagrangian function. We however showed that the minimax saddle points are not necessarily optimal value functions in learning settings. Instead, another class of the Lagrangian saddle points – the maximin-type saddle points which are derived from the *dual-form optimization* formulation – turn out to have rigorous guarantee on optimality. This observation points to new directions for this decades-old topic (see Section 5).

## 2 EPISODIC LEARNING PROCESS AND BELLMAN OPTIMAL VALUE

We start with introducing the mathematical formulation of episode learning problems, as well as the formal definitions of related concepts.

An infinite-horizon MDP is a tuple $(\mathcal{S}, \mathcal{A}, R, P, \rho_0)$, where $\mathcal{S}$ is the state space, $\mathcal{A}$ the action space, $R(s) \in [r_{\min}, r_{\max}]$ is a bounded reward (possibly negative) associated to each state $s \in \mathcal{S}$, [1] $P(s'|s, a)$ specifies action-conditioned transition probabilities between states, and $\rho_0$ is the initial distribution with $S_0 \sim \rho_0$. A MDP is *finite* if both $\mathcal{S}$ and $\mathcal{A}$ are finite sets. A policy function $\pi : \mathcal{S} \times \mathcal{A} \rightarrow [0, 1]$ specifies the action selection probabilities under each state, which induces a markov chain $\mathbf{P}_\pi[S_{t+1} = s'|S_t = s] \doteq \sum_{a \in \mathcal{A}} P(s'|s, a) \cdot \pi(a|s)$. Let $\Pi$ denote the **policy space**, i.e., the set of all policies. Without loss of generality, we assume every state $s$ in the state space $\mathcal{S}$ is reachable from the initial state under at least one policy, [2] where *reachable* means $\exists \pi \in \Pi$, $\sum_{t=0}^{\infty} \mathbf{P}_\pi[S_t = s] > 0$.

An Episodic Learning Process (ELP) White (2017); Bojun (2020) is an infinite-horizon MDP that repeatedly encounters into, and is reset by, a group of **terminal states** $\mathcal{S}_\perp$ (despite its name, a terminal state does not terminate the ELP). Formally, an infinite-horizon MDP is an ELP if there is a non-empty subset $\mathcal{S}_\perp \subseteq \mathcal{S}$ such that (1) all terminal states have homogeneous and action-agnostic outbound probabilities: $P(s'|s_1, a_1) = P(s'|s_2, a_2)$, $\forall s_1, s_2 \in \mathcal{S}_\perp$, $\forall a_1, a_2 \in \mathcal{A}$, $\forall s' \in \mathcal{S}$, (2) the initial state is a terminal state: $\rho_0(s) = 0, \forall s \notin \mathcal{S}_\perp$, and (3) the average episode length is finite under any policy: $\forall \pi \in \Pi, \mathbf{E}_\pi[T] < \infty$, where $T \doteq \inf\{t \geq 1 : S_t \in \mathcal{S}_\perp\}$ is called the **termination time**. Bojun (2020) proved that in any ELP, every policy $\pi \in \Pi$ has a unique **stationary distribution** $\rho_\pi(s, a)$ such that $\mathbf{P}_\pi[S_{t+1} = s, A_{t+1} = a] = \rho_\pi(s, a)$ if $\mathbf{P}_\pi[S_t = s, A_t = a] = \rho_\pi(s, a)$.

Following White (2017) and Bojun (2020), we use the ELP formalism to study *finite-time* decision tasks, which arguably account for most AI tasks encountered in current practice. The ELP model formulates the *learning process* of such task, which consists of an infinite number of concatenated episodes, each starting and ending at an terminal state. In general, an episode can be arbitrarily long, but will terminate in finite steps with probability $1$. The learning agent may choose to use a different behavior policy $\beta_t$ for each step $t$ of the episodic learning process (based on data collected from

---

[1] Our state-based reward formulation follows Schulman et al. (2015) and Bojun (2020), and is equivalent to action-based reward formulations in terms of expressiveness. Refer to Appendix A of Bojun (2020) for details.

[2] Unreachable states are irrelevant to the real world whatsoever, so are excluded from our treatment.

previous steps), and the goal is to have $\beta_t$ converge (as $t$ goes to infinity) towards an **optimal policy** that maximizes the expected episode-wise cumulative reward:

$$J(\pi) \doteq \mathop{\mathbf{E}}_{\zeta \sim \pi} \left[ \sum_{t=1}^{T(\zeta)} R(S_t) \right] \tag{1}$$

where $\zeta = (S_0, S_1, S_2, \dots)$ is an infinite trajectory of the markov chain induced by policy $\pi$. Note that the termination time $T$ is a random variable whose value may vary for different trajectory $\zeta$. See Section D.3 for the ELP formulation of *machine translation* as an example.

A value function $Q : \mathcal{S} \times \mathcal{A} \to \mathbb{R}$ assigns a real number to each state-action pair $(s, a)$ as the "perceived benefit" of doing action $a$ under state $s$. We call the set of all possible value functions, the **value space**, denoted by $\mathcal{Q}$. A value function $Q \in \mathcal{Q}$ is called an **optimal value** if (any of the) $Q$-**greedy policy** with $\pi_Q(a|s) > 0 \;\Rightarrow\; Q(s,a) = \max_{\bar{a}} Q(s, \bar{a})$ is an optimal policy.

Bellman optimal value is a special optimal value function that is characterized by the Bellman optimality operator. In its general form Degris et al. (2012); Sutton et al. (2011; 2014; 2016), a generalized Bellman optimality operator $\mathcal{B}^\gamma : \mathcal{Q} \to \mathcal{Q}$ transforms a value function $Q \in \mathcal{Q}$ into another value function $\mathcal{B}^\gamma Q \in \mathcal{Q}$ such that for any $(s, a) \in \mathcal{S} \times \mathcal{A}$,

$$\mathcal{B}^\gamma Q(s,a) \doteq \sum_{s' \in \mathcal{S}} P(s'|s,a) \cdot \left( R(s') + \gamma(s') \cdot \max_{a' \in \mathcal{A}} Q(s', a') \right) \tag{2}$$

where $\gamma : \mathcal{S} \to [0, 1]$ is a **discounting function** over the states.

As a classic result, if the discounting function equals a constant $\gamma_c < 1$ on every state, the corresponding Bellman optimality operator $\mathcal{B}^{\gamma_c}$ has a unique fixed point in the value space of any MDP. However, the fixed point of $\mathcal{B}^{\gamma_c}$ is generally not an optimal value with respect to the undiscounted objective (1). In the following, we present a new theorem that guarantees the uniqueness of Bellman fixed point for a more general class of discounting functions, among which a particular "episodic discounting function" gives optimal value w.r.t. objective (1). Both the uniqueness and the optimality are fundamentally rooted from an inherent graph property of ELP-MDPs. See the proof in Appendix A for detailed elaboration.

**Theorem 1.** *In any finite Episodic Learning Process $(\mathcal{S}, \mathcal{A}, P, R, \rho_0)$, let $\gamma$ be any discounting function such that $\gamma(s) < 1$ for all terminal state $s \in \mathcal{S}_\perp$, then*

 *(1) $\mathcal{B}^\gamma$ has a unique fixed point, i.e., the equation $Q = \mathcal{B}^\gamma Q$ has a unique solution.*

 *(2) The fixed point of $\mathcal{B}^\gamma$ is the limiting point of repeatedly applying $\mathcal{B}^\gamma$ to any $Q \in \mathcal{Q}$.*

 *(3) The fixed point of $\mathcal{B}^\gamma$ is an optimal value w.r.t. objective (1) if $\gamma$ is the following **episodic discounting function**:*
$$\gamma_{epi}(s) \;\doteq\; \mathbb{1}[s \notin \mathcal{S}_\perp] \;=\; \begin{cases} 1 & \text{for } s \notin \mathcal{S}_\perp \\ 0 & \text{for } s \in \mathcal{S}_\perp \end{cases}. \tag{3}$$

Since we focus on optimizing objective (1), in the rest of the paper: **Bellman optimality operator**, denoted by $\mathcal{B} \doteq \mathcal{B}^{\gamma_{epi}}$, refers to the specific operator (2) that uses the particular episodic discounting function (3); accordingly, **Bellman optimal value**, denoted by $Q^*$, refers to the unique fixed point of $\mathcal{B}$; and similarly, **Bellman optimality equation** refers to the fixed-point equation $Q = \mathcal{B}Q$ under episodic discounting, or more explicitly, refers to the following system of non-linear equations:

$$Q(s,a) = \sum_{s' \in \mathcal{S}} \max_{a' \in \mathcal{A}} P(s'|s,a) \cdot \left( R(s') + \gamma_{epi}(s') \cdot Q(s', a') \right) \quad, \forall (s,a) \in \mathcal{S} \times \mathcal{A} \tag{4}$$

It is worthwhile noting that although the Bellman optimal value function is unique, there can be many optimal value functions in an episodic learning problem. In particular, any value function that gives the same preference order with the Bellman optimality value is an optimal value function.

## 3 Lagrangian Duality and Minimax Theorem

In this section we study a minimization-based variational treatment to the Bellman optimality equation (4). We will derive a practical episodic learning algorithm based on the theory presented here, and will apply the algorithm to standard machine translation benchmarks in the next section.

Our idea is inspired by the long-known linear programming re-formulation of the closely related optimality equation for *state-value functions* $V : \mathcal{S} \to \mathbb{R}$,

$$V(s) = \max_a \sum_{s'} P(s'|s,a) \cdot \Big( R(s') + \gamma_{\text{epi}}(s') \cdot V(s') \Big) \quad , \quad \forall s. \tag{5}$$

Both the $Q$-form optimality equation (4) and the $V$-form optimality equation (5) are nonlinear due to the $\max$ operation inside, but the $V$-form equation (5) admits a *linear* re-formulation Puterman (1994):

$$\min_V \sum_{s \in \mathcal{S}} \rho_0(s) \cdot V(s) \quad \text{s.t.} \quad V(s) \geq \sum_{s'} P(s'|s,a) \cdot \Big( R(s') + \gamma_{\text{epi}}(s') \cdot V(s') \Big) \quad , \quad \forall(s,a) \tag{6}$$

Thanks to its linearity, (6) enjoys the standard LP duality properties, and in particular has *minimax equality* for its corresponding Lagrangian function, which is the basis for a recently revived thread of research on the LP approach to MDP and RL Chen and Wang (2016); Wang (2017); Cho and Wang (2017); Dai et al. (2018); Nachum and Dai (2020).

In model-free learning settings, however, the agent does not know the transition function $P$, thus cannot directly use a state-value function $V$ to compare the values of different actions (as we do not know which states an action would lead to). For this reason, most value-based algorithms focus on learning the optimal $Q$ functions Watkins (1989); Hasselt (2010); Mnih et al. (2015); Haarnoja et al. (2018). It is thus natural to ask if we can develop a variational approach directly for the $Q$-form optimality equations (4), similar to what we did to the $V$-form optimality equations.

Indeed, the $Q$-form optimality equations (4) can be similarly recast into a constrained optimization problem as follows:

$$\begin{aligned} \min_Q \quad & \mathop{\mathbf{E}}_{\zeta \sim \pi} \Big[ Q(S_T, A_T) \Big] \\ \text{s.t.} \quad & Q(s,a) \geq \sum_{s'} \max_{a'} P(s'|s,a) \cdot \Big( R(s') + \gamma_{\text{epi}}(s') \cdot Q(s',a') \Big) \quad , \quad \forall(s,a) \end{aligned} \tag{7}$$

where $\pi \in \Pi$ can be an arbitrary policy, and $T \doteq \inf\{t \geq 1 : S_t \in \mathcal{S}_\perp\}$ is the termination time. Unfortunately, unlike the $V$-form optimality equation for which the nonlinear operation $\max_a$ can be "unpacked" into $|\mathcal{A}|$ linear constraints (cf. (5) and (6)), the variational formulation (7) of the $Q$-form optimality equation is still nonlinear, due to the $\max_{a'}$ "wrapped" inside $\sum_{s'}$. As a result, although (7) can still be written into the Lagrangian form

$$\min_{Q \in \mathcal{Q}} \max_{\boldsymbol{\lambda} \geq 0} \mathop{\mathbf{E}}_{\zeta \sim \pi} \Big[ Q(S_T, A_T) \Big] + \sum_{s,a} \lambda(s,a) \cdot \Big( \mathcal{B}Q(s,a) - Q(s,a) \Big), \tag{8}$$

it is unclear if the nonlinear Q-form Lagrangian still enjoys strong duality, a key property for designing effective and principled learning algorithms Wang (2017); Dai et al. (2018).

In the following, we give an affirmative answer to this open question by proving a *minimax theorem* for the nonlinear Lagrangian of the $Q$-form Bellman optimality equation (4). As with Theorem 1, the strong Lagrangian duality is also rooted from inherent structures of the episodic learning process.

**Definition.** *Given a finite Episodic Learning Process $(\mathcal{S}, \mathcal{A}, P, R, \rho_0)$, a function $\mathcal{L}_\pi : \mathcal{Q} \times \mathbb{R}_{\geq 0}^{|\mathcal{S}| \cdot |\mathcal{A}|} \to \mathbb{R}$ is called a **Lagrangian function** with **conjugate policy** $\pi$ if*

$$\mathcal{L}_\pi(Q, \boldsymbol{\lambda}) \doteq \mathop{\mathbf{E}}_{\zeta \sim \pi} \Big[ Q(S_T, A_T) \Big] + \sum_{s,a} \lambda(s,a) \cdot \Big( \mathcal{B}Q(s,a) - Q(s,a) \Big) \tag{9}$$

*where $\pi \in \Pi$ can be an arbitrary policy, and $T \doteq \inf\{t \geq 1 : S_t \in \mathcal{S}_\perp\}$ is the termination time.*

Note that the conjugate policy $\pi$ only determines the distribution of terminal states (and actions) in the first term of $\mathcal{L}_\pi$. With any conjugate policy $\pi$, the second term of $\mathcal{L}_\pi$ always uses the Bellman *optimality* operator $\mathcal{B}$. Let us first confirm that $Q^*$ is a minimax solution of the Lagrangian function (9), as Lemma 2.1 states. The proof can be found in Appendix B.1.

**Lemma 2.1.** *In any finite ELP, for any conjugate policy $\pi$, $Q^* \in \arg\min_{Q \in \mathcal{Q}} \max_{\boldsymbol{\lambda} \geq 0} \mathcal{L}_\pi(Q, \boldsymbol{\lambda})$.*

Now, we observe that the Lagrangian $\mathcal{L}_\pi$ has an equivalent form when the Lagrangian multiplier vector is a special vector $\boldsymbol{\lambda}_\pi$ that is proportional to the *stationary distribution* of the conjugate policy $\pi$ (and proportional to the *average episode length* of $\pi$ too).

**Lemma 2.2.** *In any finite Episodic Learning Process* $(\mathcal{S}, \mathcal{A}, P, R, \rho_0)$*, for any conjugate policy* $\pi$*, let* $\mathcal{L}_\pi$ *be the corresponding Lagrangian, and let* $\boldsymbol{\lambda}_\pi$ *be the particular Lagrangian multipliers with* $\boldsymbol{\lambda}_\pi(s, a) = \rho_\pi(s, a) \cdot \mathbf{E}_{\zeta \sim \pi}[T]$*, where* $\rho_\pi$ *is the stationary distribution of* $\pi$*, then*

$$\mathcal{L}_\pi(Q, \boldsymbol{\lambda}_\pi) = J(\pi) + \sum_{s \notin \mathcal{S}_\perp} \sum_{a \in \mathcal{A}} \boldsymbol{\lambda}_\pi(s, a) \cdot \left( \max_{\bar{a}} Q(s, \bar{a}) - Q(s, a) \right) \qquad (10)$$

*Proof idea:* Applying a known *ergodic theorem* of ELP, we can transform the first term of (9) from an average over trajectories to an average over the state-action space (see Appendix B.2 for details):

$$\mathbf{E}_{\zeta \sim \pi} \left[ Q(S_T, A_T) \right] = \mathbf{E}_{\zeta \sim \pi}[T] \cdot \mathbf{E}_{S, A \sim \rho_\pi} \left[ \left( 1 - \gamma_{\text{epi}}(s) \right) \cdot Q(S, A) \right]$$

Then substituting the above equation to (9), yields

$$\begin{aligned}
\mathcal{L}_\pi(Q, \boldsymbol{\lambda}_\pi) &= \mathbf{E}_\pi[T] \cdot \mathbf{E}_{S, A \sim \rho_\pi} \left[ \mathbf{E}_{S' \sim P(\cdot | S, A)} \left[ R(S') + \gamma_{\text{epi}}(S') \cdot \max_{a'} Q(S', a') \right] - \gamma_{\text{epi}}(S) Q(S, A) \right] \\
&= \underline{\mathbf{E}_\pi[T] \cdot \mathbf{E}_{S, A \sim \rho_\pi} \left[ R(S) \right.} + \gamma_{\text{epi}}(S) \cdot \max_{a'} Q(S, a') - \gamma_{\text{epi}}(S) Q(S, A) \right] \\
&= \underline{J(\pi)} + \sum_{s \in \mathcal{S}, a \in \mathcal{A}} \mathbf{E}_\pi[T] \cdot \rho_\pi(s, a) \cdot \mathbb{1}[s \notin \mathcal{S}_\perp] \cdot \left( \max_{a'} Q(s, a') - Q(s, a) \right)
\end{aligned}$$

In above, we have again used the ergodic theorem of ELP to transform $\mathbf{E}_\pi[T] \cdot \mathbf{E}_{S \sim \rho_\pi}[R(S)]$ back to the trajectory space, so as to derive $J(\pi)$. See Appendix B.2 for the complete proof. $\square$

The first term in the dual form of the Lagrangian, i.e. in (10), is the true performance of the conjugate policy $\pi$ (in terms of objective (1)). Utilizing this fact, we can prove the strong duality property for Lagrangians conjugating with optimal policies, as the following theorem states.

**Theorem 2** (ELP Minimax Theorem). *In any finite Episodic Learning Process* $(\mathcal{S}, \mathcal{A}, P, R, \rho_0)$*, if* $\mu \in \Pi$ *is an optimal policy, then its conjugate Lagrangian* $\mathcal{L}_\mu$ *has strong duality property, with*

$$\min_{Q \in \mathcal{Q}} \max_{\boldsymbol{\lambda} \geq 0} \mathcal{L}_\mu(Q, \boldsymbol{\lambda}) = \max_{\boldsymbol{\lambda} \geq 0} \min_{Q \in \mathcal{Q}} \mathcal{L}_\mu(Q, \boldsymbol{\lambda}) = J(\mu) \qquad (11)$$

*Proof idea:* Let $\pi^*$ denote a $Q^*$-greedy policy, which is thus an optimal policy.

For any conjugate policy $\pi$, since $Q^*$ is a minimax solution of $\mathcal{L}_\pi$ (Lemma 2.1), we have $\min_Q \max_{\boldsymbol{\lambda} \geq 0} \mathcal{L}_\pi(Q, \boldsymbol{\lambda}) = \max_{\boldsymbol{\lambda} \geq 0} \mathcal{L}_\pi(Q^*, \boldsymbol{\lambda}) = \mathbf{E}_\pi[Q^*(S_T, A_T)] = \mathbf{E}_\pi[Q_{\pi^*}(S_T, A_T)]$. Because $J(\pi^*) = Q_{\pi^*}(s_\perp, a)$ for any terminal state $s_\perp \in \mathcal{S}_\perp$ and any action $a \in \mathcal{A}$ (see Appendix A.3), we have $\mathbf{E}_\pi[Q_{\pi^*}(S_T, A_T)] = J(\pi^*)$, and so $\min_Q \max_{\boldsymbol{\lambda} \geq 0} \mathcal{L}_\pi(Q, \boldsymbol{\lambda}) = J(\pi^*)$.

Again for any conjugate policy $\pi$, due to Lemma 2.2, the Lagrangian dual $\mathcal{L}_\pi(Q, \boldsymbol{\lambda}_\pi)$ for the particular multiplier $\boldsymbol{\lambda}_\pi$ attains its minimum when $Q$ achieves complementary slackness with $\boldsymbol{\lambda}_\pi$ in the second term of (10), in which case $\min_{Q \in \mathcal{Q}} \mathcal{L}_\pi(Q, \boldsymbol{\lambda}_\pi) = J(\pi)$.

So now, when the conjugate policy $\pi$ is an optimal policy $\mu$, as assumed in the theorem, we have

$$\max_{\boldsymbol{\lambda} \geq 0} \min_{Q \in \mathcal{Q}} \mathcal{L}_\mu(Q, \boldsymbol{\lambda}) \geq \min_{Q \in \mathcal{Q}} \mathcal{L}_\mu(Q, \boldsymbol{\lambda}_\mu) = J(\mu) = J(\pi^*) = \min_Q \max_{\boldsymbol{\lambda} \geq 0} \mathcal{L}_\mu(Q, \boldsymbol{\lambda}).$$

The inequality above must actually be an equality because of the universally-held *weak duality* of the Lagrangian. See Appendix B.3 for a more detailed proof. $\square$

From the proof above we can also see that $(Q^*, \pi^*)$, as the fixed point of Bellman optimality operator $\mathcal{B}$ and the corresponding $Q^*$-greedy policy (resp.), forms a *minimax saddle* point of the Lagrangian $\mathcal{L}_\mu$. In fact, we can derive general conditions for all saddle points of this kind by combining the complementary slackness conditions in (9) and (10). See the proof in Appendix B.4.

**Proposition 3.** *In any finite ELP, for any value function* $Q \in \mathcal{Q}$ *and any policy* $\pi \in \Pi$*, let* $\boldsymbol{\lambda}_\pi(s, a) = \rho_\pi(s, a) \cdot \mathbf{E}_{\zeta \sim \pi}[T]$*, then* $\mathcal{L}_\pi(Q, \bar{\boldsymbol{\lambda}}) \leq \mathcal{L}_\pi(Q, \boldsymbol{\lambda}_\pi) \leq \mathcal{L}_\pi(\bar{Q}, \boldsymbol{\lambda}_\pi)$*,* $\forall \bar{Q}, \bar{\boldsymbol{\lambda}}$*, if and only if*

*(1)* $\mathcal{B}Q(s, a) - Q(s, a) \leq 0 \qquad\qquad , \quad \forall(s, a) \in \mathcal{S} \times \mathcal{A}$

*(2)* $\rho_\pi(s, a) \cdot \left( \mathcal{B}Q(s, a) - Q(s, a) \right) = 0 \qquad , \quad \forall(s, a) \in \mathcal{S} \times \mathcal{A}$

*(3)* $\rho_\pi(s, a) \cdot \left( \max_{\bar{a}} Q(s, \bar{a}) - Q(s, a) \right) = 0 \quad , \quad \forall s \notin \mathcal{S}_\perp, a \in \mathcal{A}$

## 4 ALGORITHMIC APPLICATION TO MACHINE TRANSLATION

From last section we know that the $Q$-form Lagrangian $\mathcal{L}_\mu$ has strong duality where $\mu$ is optimal policy, and that the special Lagrangian multiplier $\boldsymbol{\lambda}_\mu$ forms a minimax saddle point (or minimax equilibrium) with the solution of (7). Formally,

$$\max_\pi J(\pi) = \min_Q \max_{\boldsymbol{\lambda} \geq 0} \mathcal{L}_\mu(Q, \boldsymbol{\lambda}) = \max_{\boldsymbol{\lambda} \geq 0} \min_Q \mathcal{L}_\mu(Q, \boldsymbol{\lambda}) = \min_Q \mathcal{L}_\mu(Q, \boldsymbol{\lambda}_\mu)$$

A simple idea to find such minimax saddle point is to minimize the rightmost side of the above equation, that is, to minimize $\mathcal{L}_\mu(Q, \boldsymbol{\lambda}_\mu)$ over the value space $\mathcal{Q}$. Although the closed form of the objective function $\mathcal{L}_\mu(Q, \boldsymbol{\lambda}_\mu)$ depends on an optimal policy $\mu$ (which might appear paradoxical at a first glance as coming up with such an optimal policy was our original goal of learning), note that estimating the gradient of $\mathcal{L}_\mu(Q, \boldsymbol{\lambda}_\mu)$ only requires knowledge about a "trace" of the optimal policy $\mu$, instead of explicit knowledge about how $\mu$ is constructed. Specifically, for parameterized value function $Q(s, a; \boldsymbol{w})$, we want to estimate $\nabla_{\boldsymbol{w}} \mathcal{L}_\mu(Q(\boldsymbol{w}), \boldsymbol{\lambda}_\mu)$, that is,

$$\nabla_{\boldsymbol{w}} \Big( \mathbf{E}_\mu[Q(S_T, A_T; \boldsymbol{w})] + \mathbf{E}_\mu[T] \underset{S,A,S' \sim \rho_\mu}{\mathbf{E}} \Big[ R(S') + \gamma_{\mathrm{epi}}(S') \max_{a'} Q(S', a'; \boldsymbol{w}) - Q(S, A; \boldsymbol{w}) \Big] \Big) \quad (12)$$

in which the optimal policy $\mu$ only plays a role in data weighting – it determines the population distribution of the terminal state-actions $(S_T, A_T)$, of the terminal time $T$, and of the transition variable $(S, A, S')$. This observation inspires a general imitation learning approach, named *LAgrangian MINimization* (LAMIN) here, in which we try to collect some demonstration data from an optimal policy, based on which we construct an estimator of the Lagrangian gradient (12), then apply standard stochastic gradient procedures to approximate the value function that minimizes $\mathcal{L}_\mu(Q(\boldsymbol{w}), \boldsymbol{\lambda}_\mu)$, which corresponds to a minimax point of the original Lagrangian function (9).

One may design different estimators of (12) to form different algorithms under the LAMIN idea. In the following we discuss two, and they both perform reasonably well in our validation experiment.

**LAMIN1**: A technical challenge of estimating (12) is to deal with the max operator in it. One idea is to relax the Lagrangian function into a smoothed version:

$$\mathcal{L}_\mu^\beta(Q(\boldsymbol{w}), \boldsymbol{\lambda}_\mu) = \mathbf{E}_\mu[Q(S_T, A_T; \boldsymbol{w})] + \mathbf{E}_\mu[T] \cdot \underset{S,A,S' \sim \rho_\mu}{\mathbf{E}} \underset{A' \sim \pi_{Q(\boldsymbol{w})}^\beta(S')}{\mathbf{E}} \Big[ \delta(S, A, S', A'; \boldsymbol{w}) \Big] \quad (13)$$

where $\pi_{Q(\boldsymbol{w})}^\beta(a|s) \doteq \frac{exp\big(Q(s,a;\boldsymbol{w})/\beta\big)}{\sum_b exp\big(Q(s,b;\boldsymbol{w})/\beta\big)}$ is the Boltzmann distribution with temperature $\beta$, and $\delta(s, a, s', a'; \boldsymbol{w}) \doteq R(s') + \gamma_{\mathrm{epi}}(s')Q(s', a'; \boldsymbol{w}) - Q(s, a; \boldsymbol{w})$ is the temporal-difference error.

The smoothed Lagrangian $\mathcal{L}_\mu^\beta$ subsumes the original Lagrangian $\mathcal{L}_\mu$ as the limiting case $\beta \to 0$, while is readily differentiable. Thus, the LAMIN1 algorithm seeks to minimize the smoothed Lagrangian $\mathcal{L}_\mu^\beta$ via SGD with unbiased gradient estimator of (13). See the pseudo-code as well as more nuanced discussion on this algorithm in Appendix D.1. Among others, it turns out that the $\beta$-smoothing trick used in LAMIN1 is more than an approximation heuristic, but may potentially serve as a practical *correction* to an inherent bias of the minimax points of the Lagrangian that we will discuss in Section 5.

**LAMIN2**: Another idea is to stick to the exact Lagrangian $\mathcal{L}_\mu$ but seek to construct "local" estimator of its gradient in a per-step manner. Specifically, let $\boldsymbol{w}_t$ be the parameter vector of $Q$ that is to be updated at a gradient step $t$, then we can estimate the value of $\nabla_{\boldsymbol{w}} \mathcal{L}_\mu(Q(\boldsymbol{w}), \boldsymbol{\lambda}_\mu)\big|_{\boldsymbol{w}=\boldsymbol{w}_t}$, for this particular paramter $\boldsymbol{w}_t$, with

$$\nabla_{\boldsymbol{w}} \Big( \mathbf{E}_\mu[Q(S_T, A_T; \boldsymbol{w})] + \mathbf{E}_\mu[T] \underset{S,A,S' \sim \rho_\mu}{\mathbf{E}} \underset{A' \sim \pi_{Q(\boldsymbol{w}^*)}(S')}{\mathbf{E}} \Big[ \delta(S, A, S', A'; \boldsymbol{w}) \Big] \Big) \Big|_{\boldsymbol{w}=\boldsymbol{w}_t} \quad (14)$$

where $\pi_{Q(\boldsymbol{w}^*)}$ is the $Q(\boldsymbol{w}^*)$-greedy policy (see Section 2). Note that the greedy policy $\pi_{Q(\boldsymbol{w}^*)}$ in (14) does not depend on $\boldsymbol{w}$ and thus is invariant to $\nabla_{\boldsymbol{w}}$; in contrast, the Boltzmann policy $\pi_{Q(\boldsymbol{w})}^\beta$ used by LAMIN1 (see (13)) will take part in the gradient computation.

The consistency of the estimator (14) is characterized by the following mathematical fact.

**Proposition 4.** *Let $\mathcal{A}$ be a finite action space, $Q(s, a; \boldsymbol{w})$ a parameterized value function. If with a given parameter vector $\boldsymbol{w}^*$, $Q(\boldsymbol{w}^*)$ suggests a unique best action $a_{\max}(s; \boldsymbol{w}^*) \doteq \arg\max_{a \in \mathcal{A}} Q(s, a; \boldsymbol{w}^*)$ for state $s$ , then*

$$\nabla_{\boldsymbol{w}} \max_{a \in \mathcal{A}} Q(s, a; \boldsymbol{w})\Big|_{\boldsymbol{w}=\boldsymbol{w}^*} = \underset{a \sim \pi_{Q(\boldsymbol{w}^*)}(s; \boldsymbol{w}^*)}{\mathbf{E}} \Big[ \nabla_{\boldsymbol{w}} Q(s, a; \boldsymbol{w}) \Big]\Big|_{\boldsymbol{w}=\boldsymbol{w}^*}$$

See the proof in Appendix D.2. In practice, the condition in Proposition 4 – i.e. $Q(\boldsymbol{w}^*)$ gives unique best actions – may be expected for large-scale continuously-valued $Q$ functions, as it is unlikely that the real-numbered values of two actions happen to be *identical* under stochastically generated $\boldsymbol{w}$.

Moreover, the local gradient estimator (14) can be combined with the $\beta$-smoothing trick by replacing the greedy policy $\pi_{Q(\boldsymbol{w}^*)}$ in (14) with Boltzmann policy $\pi^{\beta}_{Q(\boldsymbol{w}^*)}$, forming the smoothed local gradient estimator that the LAMIN2 algorithm will use. See the pseudo-code in Appendix D.2. Empirically, we found that higher temperatures help slightly improve the performance of LAMIN2.

We now apply LAMIN algorithms to Machine Translation (MT) as a case study. Learning to translate is a highly impactful AI application Wu et al. (2016), and is also an excellent example of episodic learning problem: A translation episode starts with a given sentence in a source language, and the agent takes actions to generate translation tokens one by one in a sequential manner. The episode terminates when the agent outputs a special end-of-sentence (EOS) token, at which point the translation quality is evaluated based on the source sentence and the generated translation sentence. Note that in MT, the episode length (i.e. the length of the translation sentence) is a variable controlled by the agent policy, and the episode-wise reward (i.e. the translation quality) is generally not discounted for long episodes. Just the opposite, standard MT metrics such as BLEU Papineni et al. (2002) uses *brevity penalty factor* to discount the reward of short (instead of long) translations/episodes. See Appendix D.3 for the exact episodic learning formulation of MT.

Given a source sentence $X$, most MT metrics measure the quality of a generated translation $Y$ by comparing the similarity between $Y$ and some *reference translation* $Z$ of $X$, where $Z$ is typically provided by human expert given $X$. It follows that the translation policy used by the human expert (which maps $X$ to $Z$) must be an optimal policy under such metric. Importantly, a trace of the optimal translation policy, in the form of a collection of source-reference sentence pairs, is indeed widely available in standard MT benchmarks, which can be readily used to power LAMIN algorithms. In general, the same idea applies to all machine learning problems with *imitation reward* where the performance metric is based on similarity comparison to reference/ground-truth outputs.

We tested our algorithmic ideas using the WMT English→German (en2de) dataset, one of the most influential MT benchmarks. We parameterized value function $Q$ with the standard TransformerBase neural network Vaswani et al. (2017), and trained the Transformer-based Q-function using LAMIN1 and LAMIN2, with varying temperature $\beta$, then tested the Q-greedy policy with standard BLEU metric using the WMT'14 NewsTest data. See Appendix D.4 for details in experimentation setup.

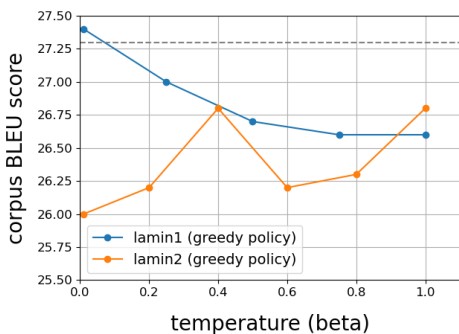

Figure 1: Corpus BLEU scores of LAMIN1 and LAMIN2 under different Boltzmann temperature $\beta$. Greedy policy is used in both cases. The gray dashed line indicates the baseline performance of 27.3 achieved by a beam-search policy in Vaswani et al. (2017).

Figure 2: Learning curves of some LAMIN variants. Beam search with a beam size of 4 is used for the orange line, which matches the same setting with the baseline result of 27.3 (gray dashed line). Other lines use greedy policy which is about 2x faster.

Figure 1 illustrates the performance of LAMIN1 and LAMIN2 under a spectrum of Boltzmann temperatures, from 0.01 (in which case the algorithm is effectively minimizing the exact Lagrangian function) to 1.0 (which corresponds to the softmax distribution). Table 1 and 2 in Appendix D.4 reports the numerical scores. We see that LAMIN1 has slightly better performance than LAMIN2

under most temperatures. Interesting, higher temperatures tend to help the performance of LAMIN2 but hurt that of LAMIN1 (albeit with limited margins in both cases).

Notably, LAMIN1 with $\beta = 0.01$ attains a BLEU score of $27.4$ with a *greedy policy*. In comparison, when standard supervised learning is used, the same neural network famously attains $27.3$ only if further combined with a systematic *beam search* Vaswani et al. (2017). So, our algorithm achieves comparable performance with the state-of-the-art result *without* time-consuming search procedure at decision/translation time. On the other hand, when incorporating with the same beam search procedure, the value function trained by LAMIN1 (with $\beta = 0.01$) attains $28.7$, or $1.4$ BLEU score higher. Figure 2 shows the learning curves of the algorithm, with and without beam search.

## 5 MINIMAX VALUES VS MAXIMIN VALUES

The Lagrangian method we explored so far has focused on a particular class of Lagrangian saddle points, the minimax saddle points $\arg\min_Q \max_\lambda \mathcal{L}_\pi$, which correspond to exactly the set of optimal solutions for the minimization-form (or *primal-form*) variational problem (7). Let's call such a value function, a **minimax value function**. It seems that existing research on the Lagrangian method have been exclusively focusing on studying minimax value functions (or minimax points of the Lagrangian); see Puterman (1994); Cho and Wang (2017); Dai et al. (2018); Nachum and Dai (2020) for examples. Indeed, as Lemma 2.1 confirmed, the Bellman-optimal value $Q^*$, as an optimal value function, is a minimax value function. However, a minimax value function, or equivalently, a minimax solution of $\mathcal{L}_\pi$ (for any $\pi$), is *not* necessarily an optimal value function.

Figure 3 shows an example, where state $4$ and $5$ are terminal states, from which any action leads to state $0$. Choosing action $1, 2, 3$ under state $0$ deterministically transits to state $1, 2, 3$, respectively. All actions under state $1$ lead to state $4$, and all actions under state $2$ and $3$ lead to state $5$. The agent only receives non-zero rewards at terminal states, with $R(4) = 1$, $R(5) = 2$. The initial state is set to state $4$. Clearly, an optimal policy of this ELP should only choose action $2$ or $3$ (or both), but not action $1$, under state $0$.

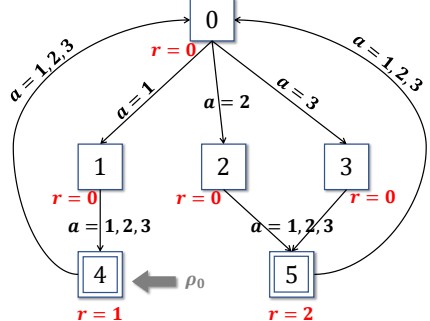

Figure 3: An ELP example

One can verify that for this ELP, the constant value function $Q_{\min}(s, a) \equiv 2$ is a minimax value function. In particular, we have $2 = Q_{\min}(0, 1) = 0 + 1 \cdot Q_{\min}(1, a) > 1 + 0 \cdot Q_{\min}(4, a) = 1$, which does satisfy the constraints of (7) (that $Q \geq \mathcal{B}Q$). But the constant value function $Q_{\min}$ is certainly not optimal as it assigns the same value to all actions under state $0$. Moreover, because $Q_{\min}(1, a) > \mathcal{B}Q_{\min}(1, a) = 1$ for all $a$, the $\lambda$ corresponding to $Q_{\min}$ needs to have $\lambda(1, a) = 0$ for all $a$. Such a $\lambda$ cannot encode any policy.

The problem with minimax value functions is that they only guarantee optimality for optimal actions, but do not enforce sub-optimality for sub-optimal actions (such as the action $1$ under state $0$). This inadequacy of minimax value functions may explain our experimental observation in last section that adding a softmax temperature in the Lagrangian leads to better value functions (because the softmax averaging motivates, to some extent, the optimizer to downgrade sub-optimal actions to further lower the Lagrangian). Moreover, we remark that this is not only a problem for $Q$-functions or for episodic discounting, but the minimax solutions of the more popular $V$-form Lagrangian in discounted-MDPs Dai et al. (2018) suffer from the same issue too (see Appendix C.4 for a counter-example for the minimax $V$-functions in discounted setting).

Interestingly enough, it turns out that value optimality is indeed guaranteed for another class of saddle points of the Lagrangian – the *maximin saddle points*. Specifically, with the same argument as in Lemma 2.1 one can show that $Q^*$, the fixed point of $\mathcal{B}$, is also the optimal solution of the following "mirrored" problem to (7). See Appendix C.1 for the proof.

**Lemma 5.1.** *In any finite ELP, $Q^*$ is an optimal solution of*

$$\max_Q \ \mathbb{E}_{\zeta \sim \pi} \left[ Q(S_T, A_T) \right] \quad s.t. \quad Q(s, a) \leq \mathcal{B}Q(s, a) \ , \ \forall (s, a) \tag{15}$$

*for any conjugate policy $\pi \in \Pi$. Equivalently, $Q^* \in \arg\max\limits_{Q \in \mathcal{Q}} \min\limits_{\boldsymbol{\lambda} \geq 0} \mathcal{L}_\pi(Q, \boldsymbol{\lambda})$.*

We call an optimal solution of (15), a **maximin value function** (with respect to the Lagrangian $\mathcal{L}_\pi$). Different from minimax value functions, a maximin value function enforces sub-optimality for all (truly) sub-optimal actions, *and* at the same time can guarantee optimality for at least one (truly) optimal action. Consequently, a maximin value function always and only induces optimal policy.

**Theorem 5.** *In any finite ELP, for any conjugate policy $\pi$, let $Q_{\max}$ be an maximin value function of the Lagrangian $\mathcal{L}_\pi$ – i.e. let $Q_{\max}$ be an optimal solution of (15) – then $Q_{\max}$ is an optimal value function, in the sense that $Q_{\max}$-greedy policy maximizes the episodic-reward objective (1).*

*Proof idea:* First observe that $Q_{\max}(s, a) \leq Q^*$ at all $(s, a)$. This is because $Q_{\max}$, as a feasible solution of (15), has $Q_{\max} \leq \mathcal{B}Q_{\max} \leq \mathcal{B}\mathcal{B}Q_{\max} \cdots \leq Q^*$. Now $Q_{\max} \leq Q^*$ implies that for all actions sub-optimal to $Q^*$ (under a state), they can only have even lower values in $Q_{\max}$. So, it is enough to prove that $\max_a Q_{\max}(s, a) = \max_a Q^*(s, a)$ at every non-terminal state $s$ that is reachable by $Q_{\max}$-greedy policy – in that case the footprint of a $Q_{\max}$-greedy policy will be a subset of the footprint of a $Q^*$-greedy policy, in which only $Q^*$-optimal actions are selected. Note that $Q_{\max}$'s values on terminal states do not affect its footprint, due to the ELP conditions.

For an arbitrary state-action pair $(s, a)$, let $(\mathcal{S} \times \mathcal{A})_{\text{next}}$ denote the set of all the non-terminal $(s', a')$ pairs that can directly follow $(s, a)$ under the $Q_{\max}$-greedy policy. We can prove that

$$Q_{\max}(s, a) = Q^*(s, a)$$
$$\Rightarrow Q_{\max}(s', a') = Q^*(s', a') \quad \text{and} \quad \max_{\bar{a}} Q_{\max}(s', \bar{a}) = \max_{\bar{a}} Q^*(s', \bar{a}) \ , \ \forall(s', a') \in (\mathcal{S} \times \mathcal{A})_{\text{next}}$$

which would enable a proof-by-induction, starting from some terminal state where $Q_{\max}$ and $Q^*$ equal to each other (such terminal state exists because both $Q_{\max}$ and $Q^*$ are optimal solutions of (15)). See Appendix C.2 for the complete proof. $\square$

In light of the symmetry breaking between the minimax-type and the maximin-type Lagrangian saddle points (in terms of optimality), we believe the latter, as well as minimax value functions and the corresponding *dual-form* variational problem (15), deserve more attentions from the community.

## 6 RELATED WORKS

A main challenge of this work comes from the non-linearity in both the Bellman optimality equation and the associated $Q$-form Lagrangian being studied. In contrast, most related research focused on linear treatments to related objects. For a linear *policy-specific* Bellman operator $\mathcal{B}_\pi$ (which replaces $\max_a$ with a linear average), White (2017) proved that $\mathcal{B}_\pi$ has unique fixed point in episodic learning setting. The linear operator $\mathcal{B}_\pi$ leads to a LP (re-)formulation, based on which the "DICE" family of policy evaluation algorithms were developed Nachum et al. (2019); Yang et al. (2020). On the policy optimization side, an active thread of research used saddle-point optimization to solve the linear $V$-form Lagrangian, again relying on the generic LP duality inherited from the linear treatment Chen and Wang (2016); Wang (2017); Cho and Wang (2017); Dai et al. (2018); Chen et al. (2018); Serrano and Neu (2020); Si et al. (2004). The underlying techniques in these linear settings are not directly applicable to our nonlinear treatment in this work.

The disparity between discounted formalism and episodic learning practice is well recognized in reinforcement learning literature. The RL textbook Sutton and Barto (2018) devoted its Section 10.4 to the issue of deprecating the discounted formalism. The DP textbook Bertsekas and Tsitsiklis (1996) subsumed the discounted setting as a special case of an finite-termination setting. A special case of Theorem 1 dedicated to episodic discounting was proved by Bertsekas and Tsitsiklis (1991).

The WMT machine translation benchmark used in this paper is a fruitful driver of the rapid technical advances in Neural Machine Translation in recent years Wu et al. (2016); Koehn and Knowles (2017); Vaswani et al. (2017). MDP-based techniques have been actively studied as a promising method for this problem Ranzato et al. (2016); Edunov et al. (2018); Bahdanau et al. (2017); Wu et al. (2018), but with relatively limited effectiveness observed so far Choshen et al. (2020). To our best knowledge, the LAMIN algorithm is one of the first MDP-based solutions that is able to train Transformer-scale neural networks *independently* (without pretraining or ensemble learning with the aid of other techniques) to attain competitive performance on the WMT benchmark.

## 7 REPRODUCIBILITY STATEMENT

Every theorem, lemma, proposition in the main text or in the appendix of this paper has been given complete proof (unless already proved in other papers in which case the reference was given). For the experiment part, detailed experimentation setting was described in Appendix D.4. Experimentation code was attached in Supplementary Material.

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

# A PROOFS OF BELLMAN OPTIMALITY

In this section we prove the uniqueness and optimality of the solution of generalized Bellman equation in episodic learning setting.

## A.1 PROPERTIES OF EPISODIC LEARNING PROCESS

This section presents a series of mathematical properties of ELP; the first three is known, the rest are new. Most of our theoretical results in this paper are based on these properties.

**Proposition 6.** *For any policy $\pi$ in any ELP, the Markov chain induced by policy $\pi$ is irreducible: $\forall s, s' \in \mathcal{S}_\pi, \sum_{\tau=1}^{\infty} \mathbf{P}_\pi[S_{t+\tau} = s' | S_t = s] > 0$. (Bojun (2020), Lemma 1.1)*

**Proposition 7.** *For any policy $\pi$ in any ELP, the Markov chain induced by policy $\pi$ is positive recurrent: $\forall s \in \mathcal{S}_\pi, \mathbf{E}_\pi[T_s] < \infty$, where $T_s$ is the recurrent time of $s$. (Bojun (2020), Lemma 1.2)*

**Proposition 8.** *For any policy $\pi$ in any ELP, let $f : \mathcal{S} \to \mathbb{R}$ be a real-valued function over the states, we have (Bojun (2020), Theorem 4)*

$$\mathbf{E}_{S \sim \rho_\pi} \left[ f(S) \right] = \mathbf{E}_{\zeta \sim \pi} \left[ \sum_{t=1}^{T} f(S_t) \right] / \mathbf{E}_{\zeta \sim \pi} \left[ T \right] \tag{16}$$

We write $Q_1 \geq Q_2$ for two value functions $Q_1, Q_2 \in \mathcal{Q}$ iff $Q_1(s, a) \geq Q_2(s, a), \forall (s, a) \in \mathcal{S} \times \mathcal{A}$. First observe that the generalized Bellman optimality operator (2) is *monotonic*.

**Proposition 9.** *For any discounting function $\gamma : \mathcal{S} \to [0, 1]$, the generalized Bellman optimality operator $\mathcal{B}^\gamma$ is a monotonic operator: $Q_1 \geq Q_2 \Rightarrow \mathcal{B}^\gamma Q_1 \geq \mathcal{B}^\gamma Q_2$, for all $Q_1, Q_2 \in \mathcal{Q}$.*

*Proof.* Rewrite (2) as $\mathcal{B}^\gamma Q(s, a) = \sum_{s'} \left( P(s'|s, a) \cdot \gamma(s') \right) \cdot \max_{a'} Q(s', a') + \sum_{s'} P(s'|s, a) R(s')$. As $Q_1(s', a') \geq Q_2(s', a')$ for all $(s', a')$, we have $\max_{a'} Q_1(s', a') \geq \max_{a'} Q_2(s', a')$ for all $s'$, and thus $\sum_{s'} \left( P(s'|s, a) \cdot \gamma(s') \right) \cdot \max_{a'} Q_1(s', a') \geq \sum_{s'} \left( P(s'|s, a) \cdot \gamma(s') \right) \cdot \max_{a'} Q_2(s', a')$ because $P(s'|s, a) \cdot \gamma(s') \geq 0$ for all $s'$. $\square$

Now, as a well-known special case, if the discounting function is a constant less than 1, the corresponding Bellman optimality operator is a *contraction mapping* with respect to the maximum-norm distance over the value space $\mathcal{Q}$, which guarantees, by the Banach fixed-point theorem, that there is a special value function $Q_c^*$ which is both the unique fixed point ($Q_c^* = \mathcal{B}_c Q_c^*$) and the unique limiting point of the Bellman optimality operator ($Q_c^* = \lim_{n \to \infty} (\mathcal{B}_c)^n Q$, $\forall Q \in \mathcal{Q}$).

Unfortunately, $\mathcal{B}^\gamma$ loses the above contraction property under general discounting.

**Proposition 10.** *In any ELP where there is a single state $s^*$ and a single action $a^*$ such that doing $a^*$ under $s^*$ only goes to non-terminal states $s'$, i.e. $\sum_{s' \notin \mathcal{S}_\perp} P(s'|s^*, a^*) = 1$, the Bellman operator $\mathcal{B}$ with episodic discounting (3) is **not** a contraction mapping with respect to the maximum-norm distance: For some $Q_1, Q_2 \in \mathcal{Q}$, $\max_{s,a} |Q_1(s, a) - Q_2(s, a)| = \max_{s,a} |\mathcal{B}Q_1(s, a) - \mathcal{B}Q(s, a)|$.*

*Proof.* Consider two value functions with constant difference everywhere: $Q_1(s, a) \equiv Q_2(s, a) + \delta$, for some $\delta > 0$. Clearly, $\max_{s,a} |Q_1(s, a) - Q_2(s, a)| = \delta$. On the other hand, for any $(s, a)$ we have $|\mathcal{B}Q_1(s, a) - \mathcal{B}Q_2(s, a)| = \sum_{s' \in \mathcal{S}} P(s'|s, a) \cdot \gamma(s') \cdot \delta \leq \delta$, which equals $\delta$ at the particular $(s^*, a^*)$ because $\sum_{s' \in \mathcal{S}} P(s'|s^*, a^*) \cdot \gamma(s') \cdot \delta = \sum_{s' \notin \mathcal{S}_\perp} P(s'|s^*, a^*) \cdot 1 \cdot \delta = \delta$. $\square$

Proposition 10 seems to be a negative result for most non-trivial ELPs – it says $\mathcal{B}$ is not a contraction mapping unless we always has a chance to immediately terminate an episode no matter where we are (even when we are at terminal states, at which point the episode has not effectively started yet).

But interestingly, it turns out that the episodic Bellman optimality operator $\mathcal{B}$ still has unique fixed and limiting point in *all* finite ELPs, not because of the contraction property as in discounted-MDPs, but because of a graph property dedicated to the family of ELPs, as will be shown next.

**Lemma 2.1.** *In any Episodic Learning Process* $(\mathcal{S}, \mathcal{A}, P, R, \rho_0)$, *for any subset of states* $\Omega \subseteq \mathcal{S}$ *and any policy* $\pi$, *let* $\mathcal{C}_\pi(\Omega) \doteq \{ s' : \exists s \in \Omega, \mathbf{P}_\pi[S_{t+1} = s' | S_t = s] > 0 \}$ *be the set of all successor states that are one-step reachable from* $\Omega$ *under* $\pi$, *and let* $\mathcal{S}_\pi$ *be the set of states that are ever reachable under* $\pi$ *(from initial states, in finite steps), then* $\mathcal{C}_\pi(\Omega) \subseteq \Omega$ *only if* $\mathcal{S}_\pi \subseteq \Omega$.

*Proof.* For contradiction suppose $\mathcal{C}_\pi(\Omega) \subseteq \Omega$ and yet there is a $\pi$-reachable state $s^* \in \mathcal{S}_\pi$ that is outside the given subset $\Omega$. We will show that in this case, it is possible to construct a policy $\mu$ (that is possibly different from $\pi$) such that $s^*$ is also reachable under $\mu$, and that $\mu$ admits an infinite trajectory that passes through $s^*$ and never return back to $s^*$ (see Figure 4 below). This would contradict with Proposition 7 above which asserts that a $\mu$-reachable state $s^*$ must have finite mean recurrence time under $\mu$.

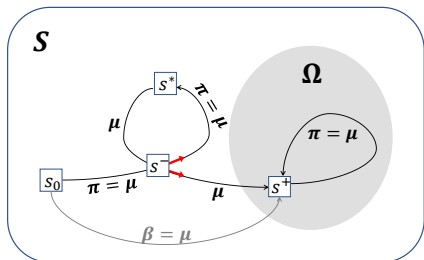

Figure 4: An $\mu$-admissible trajectory that goes through $s^*$ but never returns.

Specifically, first observe that $\mathcal{C}_\pi(\Omega) \subseteq \Omega$ means $\Omega$ is a absorbing subset under $\pi$ so that once we get into $\Omega$ we would never get out.

Then observe that the existence of a $\pi$-reachable state $s^* \notin \Omega$ entails that $\Omega$ cannot contain all initial states, as otherwise from any of the initial states (in $\Omega$) we cannot go outside the absorbing subset $\Omega$ to reach $s^*$. Let $s_0$ be such an initial state that is outside $\Omega$, from which we can reach $s^*$ under $\pi$ (as assumed) without reaching any state in $\Omega$ in the middle (otherwise we never reach $s^*$).

Now, pick an arbitrary state $s^+$ in $\Omega$, there must be some policy $\beta$ under which we can reach $s^+$ from $s_0$ (as states unreachable under any policy should not show up in $\mathcal{S}$ in the first place, see Section 2). Without loss of generality we can again assume that we can reach $s^+$ from $s_0$ under $\beta$ without going through any other state in $\Omega$ in the middle because otherwise we can simply re-define $s^+$ to be the first state in $\Omega$ that we have encountered on the path (from $s_0$ to the "old $s^+$").

So far we have obtained an initial state $s_0$ outside $\Omega$, from which there is an admissible path $s_0 \xrightarrow{\pi} s^*$ for policy $\pi$, and an admissible path $s_0 \xrightarrow{\beta} s^+$ for policy $\beta$. Both paths only contain states outside $\Omega$ (except $s^+$). Now we construct policy $\mu$ as follows: we ask $\mu$ to copy $\pi$ on states in the path $s_0 \xrightarrow{\pi} s^*$, and ask $\mu$ to copy $\beta$ on states in the path $s_0 \xrightarrow{\beta} s^+$. If a state shows up in both paths, we ask $\mu$ to be a (probability) mixture of both $\pi$ and $\beta$ on that state. Clearly, we can reach both $s^*$ and $s^+$ from $s_0$ under $\mu$ (note that the policy mixing only decreases the probabilities to reach $s^*$ and $s^+$ but does not change their reachability).

By Proposition 6, since both $s^*$ and $s^+$ are in $\mathcal{S}_\mu$, we must be able to go from $s^*$ to $s^+$ under $\mu$. Note that so far we have only prescribed $\mu$'s behavior *outside* $\Omega$. Our final step is to ask $\mu$ to copy $\pi$ for all states in $\Omega$, so that $\Omega$ is also an absorbing subset for $\mu$, meaning that once we reach $s^+$ (from $s^*$), we will be stuck in $\Omega$ without going back to $s^*$ (which is outside $\Omega$ by our assumption at the beginning of the proof). In this way, we have constructed a policy $\mu$, under which we can first go from $s_0$ to $s^*$, then go from $s^*$ to $s^+$, and then be stuck in $\Omega$ forever without returning to $s^*$. A possibility of such an infinite trajectory under $\mu$ directly contradicts with Proposition 7. □

### A.2 PROOF OF THEOREM 1 (1) (2)

Lemma 2.1 says that in ELPs, any absorbing subset of $\mathcal{S}$ under a policy must contain all reachable states of this policy. In particular, there cannot exist an absorbing subset *outside* the reachable set $\mathcal{S}_\pi$. Utilizing this fact, we can prove the first two statements of Theorem 1.

(**Theorem 1 (1) (2)**). *In any ELP* $(\mathcal{S}, \mathcal{A}, P, R, \rho_0)$ *with finite state space* $\mathcal{S}$ *and finite action space* $\mathcal{A}$, *let* $\gamma$ *be any discounting function such that* $\gamma(s) < 1$ *for all terminal state* $s \in \mathcal{S}_\perp$, *then*

(1) $\mathcal{B}^\gamma$ *has a unique fixed point, i.e., the equation* $Q = \mathcal{B}^\gamma Q$ *has a unique solution.*

(2) *The fixed point of* $\mathcal{B}^\gamma$ *is also the limiting point of repeatedly applying* $\mathcal{B}^\gamma$ *to any* $Q \in \mathcal{Q}$.

*Proof.* For any two value functions $Q_1$ and $Q_2$, consider their $L_\infty$-distance

$$d(Q_1, Q_2) \doteq \max_{s \in \mathcal{S}} \; d_s(Q_1, Q_2)$$

where

$$d_s(Q_1, Q_2) \doteq \max_{a \in \mathcal{A}} \; |Q_1(s, a) - Q_2(s, a)|.$$

As usual, we have

$$d(\mathcal{B}^\gamma Q_1, \mathcal{B}^\gamma Q_2) = \max_{s \in \mathcal{S}} \max_{a \in \mathcal{A}} \Big| \sum_{s' \in \mathcal{S}} P(s'|s, a) \cdot \gamma(s') \cdot \Big( \max_{a_1'} Q_1(s', a_1') - \max_{a_2'} Q_2(s', a_2') \Big) \Big|$$

$$\leq \max_{s \in \mathcal{S}} \max_{a \in \mathcal{A}} \sum_{s' \in \mathcal{S}} P(s'|s, a) \cdot \gamma(s') \cdot \Big| \max_{a_1'} Q_1(s', a_1') - \max_{a_2'} Q_2(s', a_2') \Big|$$

$$\leq \max_{s \in \mathcal{S}} \max_{a \in \mathcal{A}} \sum_{s' \in \mathcal{S}} P(s'|s, a) \cdot \gamma(s') \cdot \max_{a'} \Big| Q_1(s', a') - Q_2(s', a') \Big|$$

$$= \max_{s \in \mathcal{S}} \max_{a \in \mathcal{A}} \sum_{s' \in \mathcal{S}} P(s'|s, a) \cdot \gamma(s') \cdot d_{s'}(Q_1, Q_2). \tag{17}$$

Traditionally, it was assumed that $\gamma(s') \equiv \gamma_c < 1$ for all states, thus the $\gamma(s')$ term in (17) can be readily moved out of the sum, immediately yielding $d(\mathcal{B} Q_1, \mathcal{B} Q_2) \leq \gamma_c \cdot d(Q_1, Q_2)$. When $\gamma(s')$ is not constant and is allowed to be 1 for non-terminal states, applying the operator $\mathcal{B}^\gamma$ to $Q_1$ and $Q_2$ cannot guarantee to reduce $d(Q_1, Q_2)$, as discussed in Proposition 10.

However, by utilizing the graph property as proved in Lemma 2.1, we can show that $\mathcal{B}^\gamma$ guarantees to reduce the per-state distance $d_s(Q_1, Q_2)$ at some "support dimension" $s$, so that if we repeatedly apply $\mathcal{B}^\gamma$, the set of "support states" will become smaller and smaller and eventually become empty at which point the overall $L_\infty$-distance gets reduced (by the composite operator of repeatedly applying $\mathcal{B}^\gamma$).

Specifically, for given $Q_1, Q_2 \in \mathcal{Q}$, we will identify a sequence of *proper* subsets of states

$$\mathcal{S} = \texttt{d-support}(0) \supset \texttt{d-support}(1) \supset \texttt{d-support}(2) \supset \cdots \supset \texttt{d-support}(|\mathcal{S}|) \tag{18}$$

such that

$$s \notin \texttt{d-support}(k) \;\Rightarrow\; \forall i \geq k, \; d_s\Big( (\mathcal{B}^\gamma)^i Q_1, (\mathcal{B}^\gamma)^i Q_2 \Big) < d(Q_1, Q_2) \tag{19}$$

for all $k \geq 0$. The construction of the subsets is by induction, and is based on the following insight:

**Lemma 2.2.** *Under the condition of Theorem 1 , if* (19) *holds for* $k - 1$, *then*

$$\exists s^* \in \texttt{d-support}(k), \;\; \text{such that} \;\; \forall i \geq k, \; d_{s^*}\Big( (\mathcal{B}^\gamma)^i Q_1, (\mathcal{B}^\gamma)^i Q_2 \Big) < d(Q_1, Q_2)$$

*Proof.* We first refactor (17) a little bit, which actually holds in a per-state sense, so

$$d_s(\mathcal{B}^\gamma Q_1, \mathcal{B}^\gamma Q_2) \leq \max_{a \in \mathcal{A}} \sum_{s' \in \mathcal{S}} P(s'|s, a) \cdot \gamma(s') \cdot d_{s'}(Q_1, Q_2)$$

$$\leq \max_{s' \in \mathcal{S}} d_{s'}(Q_1, Q_2) \; = \; d(Q_1, Q_2). \tag{20}$$

Recursively applying (20) gives

$$d_s\Big( (\mathcal{B}^\gamma)^k Q_1, (\mathcal{B}^\gamma)^k Q_2 \Big) \leq \max_{a \in \mathcal{A}} \sum_{s' \in \mathcal{S}} P(s'|s, a) \cdot \gamma(s') \cdot d_{s'}\Big( (\mathcal{B}^\gamma)^{k-1} Q_1, (\mathcal{B}^\gamma)^{k-1} Q_2 \Big)$$

$$= \sum_{s' \in \mathcal{S}} P(s'|s, a_{\max}(s)) \cdot \gamma(s') \cdot d_{s'}\Big( (\mathcal{B}^\gamma)^{k-1} Q_1, (\mathcal{B}^\gamma)^{k-1} Q_2 \Big) \tag{21}$$

$$\leq \sum_{s' \in \mathcal{S}} P(s'|s, a_{\max}(s)) \cdot \gamma(s') \cdot d(Q_1, Q_2) \leq d(Q_1, Q_2)$$

where $a_{\max}(s) \doteq \arg\max_a \sum_{s'\in\mathcal{S}} P(s'|s,a)\cdot\gamma(s')\cdot d_{s'}\big((\mathcal{B}^\gamma)^{k-1}Q_1,(\mathcal{B}^\gamma)^{k-1}Q_2\big)$.

Now we prove that the inequality (21) must be strict for some support-state $s^* \in$ d-support$(k)$. In particular, we will prove

$$\sum_{s'\in\mathcal{S}} P(s'|s^*,a_{\max}(s^*))\cdot\gamma(s')\cdot d_{s'}\big((\mathcal{B}^\gamma)^{k-1}Q_1,(\mathcal{B}^\gamma)^{k-1}Q_2\big) < d(Q_1,Q_2) \qquad (22)$$

**Case** 1: There is an $s^* \in$ d-support$(k)$ with $\sum_{s'\in\text{d-support}(k)} P\big(s'|s^*,a_{\max}(s^*)\big) < 1$. In this case it's possible to go from $s^*$ to some $s'$ outside the subset of d-support$(k)$. For such $s'$ we have $\gamma(s')\cdot d_{s'}\big((\mathcal{B}^\gamma)^{k-1}Q_1,(\mathcal{B}^\gamma)^{k-1}Q_2\big) < d(Q_1,Q_2)$ (because (19) holds for $k-1$ as assumed), thus

$$\sum_{s'\in\mathcal{S}} P(s'|s^*,a_{\max})\cdot\gamma(s')\cdot d_{s'}\big((\mathcal{B}^\gamma)^{k-1}Q_1,(\mathcal{B}^\gamma)^{k-1}Q_2\big)$$

$$< \sum_{s'\notin\text{d-support}(k)} P(s'|s^*,a_{\max})\cdot d(Q_1,Q_2) \;+$$

$$\sum_{s'\in\text{d-support}(k)} P(s'|s^*,a_{\max})\cdot\gamma(s')\cdot d_{s'}\big((\mathcal{B}^\gamma)^{k-1}Q_1,(\mathcal{B}^\gamma)^{k-1}Q_2\big)$$

$$\leq \sum_{s'\notin\text{d-support}(k)} P(s'|s^*,a_{\max})\cdot d(Q_1,Q_2) + \sum_{s'\in\text{d-support}(k)} P(s'|s^*,a_{\max})\cdot d(Q_1,Q_2)$$

$$= d(Q_1,Q_2).$$

**Case** 2: For all $s \in$ d-support$(Q_1,Q_2)$, $\sum_{s'\in\text{d-support}(Q_1,Q_2)} P\big(s'|s,a_{\max}(s)\big) = 1$. This is equivalent to say that there exists a policy – which would choose $a_{\max}(s)$ under the corresponding $s$ – such that it is *impossible* to move from any state in d-support$(k)$ to a state outside d-support$(k)$ under this policy. In other words, let $\mu$ be such a policy, we have

$$\mathcal{C}_\mu\big(\text{d-support}(k)\big) \subseteq \text{d-support}(k)$$

which, by Lemma 2.1, entails

$$\mathcal{S}_\mu \subseteq \text{d-support}(k), \qquad (23)$$

(23) literally says that the set of support-states d-support$(k)$, as an absorbing subset under $\mu$ as assumed in case 2, must contain all reachable states under $\mu$. By the definition of ELP, these $\mu$-reachable states must in turn contain at least one terminal state (otherwise we would not have finite episode under $\mu$ at all). Let $s_\perp \in \mathcal{S}_\mu \subseteq$ d-support$(k)$ be such a reachable terminal state under $\mu$. Since $s_\perp$ is reachable under $\mu$ (and $\mu$ chooses $a_{\max}(s)$ under each $s$), there must also be an $s^* \in \mathcal{S}_\mu$ such that $P(s_\perp|s^*,a_{\max}(s^*)) > 0$. Because $\gamma(s_\perp) < 1$ as assumed as the general condition of Theorem 1, we have

$$\sum_{s'\in\mathcal{S}} P(s'|s^*,a_{\max})\cdot\gamma(s')\cdot d_{s'}\big((\mathcal{B}^\gamma)^{k-1}Q_1,(\mathcal{B}^\gamma)^{k-1}Q_2\big)$$

$$\leq \sum_{s'\in\mathcal{S}} P(s'|s^*,a_{\max})\cdot\gamma(s')\cdot d(Q_1,Q_2)$$

$$= \Big( P(s_\perp|s^*,a_{\max})\cdot\gamma(s_\perp) + \sum_{s'\in\mathcal{S}\setminus\{s_\perp\}} P(s'|s^*,a_{\max})\cdot\gamma(s') \Big)\cdot d(Q_1,Q_2)$$

$$< \Big( P(s_\perp|s^*,a_{\max}) + \sum_{s'\in\mathcal{S}\setminus\{s_\perp\}} P(s'|s^*,a_{\max})\cdot\gamma(s') \Big)\cdot d(Q_1,Q_2)$$

$$\leq \Big( P(s_\perp|s^*,a_{\max}) + \sum_{s'\in\mathcal{S}\setminus\{s_\perp\}} P(s'|s^*,a_{\max}) \Big)\cdot d(Q_1,Q_2)$$

$$= d(Q_1,Q_2).$$

Now we have proved that $\exists s^* \in \texttt{d-support}(k)$, $d_{s^*}\left((\mathcal{B}^\gamma)^k Q_1, (\mathcal{B}^\gamma)^k Q_2\right) < d(Q_1, Q_2)$. It is straightforward to verify that the same proof idea applies to all $i > k$ too (in case 1, we still have $\gamma(s') \cdot d_{s'}\left((\mathcal{B}^\gamma)^{i-1} Q_1, (\mathcal{B}^\gamma)^{i-1} Q_2\right) < d(Q_1, Q_2)$ because (19) holds for $k-1$; in case 2, the existence of $s_\perp$ is a graph property that is independent of how many times $\mathcal{B}^\gamma$ is applied). $\qquad\square$

With Lemma 2.2, we can construct each subset $\texttt{d-support}(k)$ in (18) by removing the $s^*$ from $\texttt{d-support}(k-1)$. It's clear that (19) will hold for the sequence of support subsets thus constructed. In particular, note that Lemma 2.2 holds for $k = 0$ without the inductive condition (that (19) holds for $k-1$) because in this case it's impossible to go outside $\texttt{d-support}(0) = \mathcal{S}$ as in Case 1, so only Case 2 is possible (and in this case the proof does not need the inductive condition).

Now, (18) implies that $\texttt{d-support}(|\mathcal{S}|)$ is empty set. Substituting this observation into (19), yields $d_s\left((\mathcal{B}^\gamma)^{|\mathcal{S}|} Q_1, (\mathcal{B}^\gamma)^{|\mathcal{S}|} Q_2\right) < d(Q_1, Q_2)$ for all $s \in \mathcal{S}$, which means $d\left((\mathcal{B}^\gamma)^{|\mathcal{S}|} Q_1, (\mathcal{B}^\gamma)^{|\mathcal{S}|} Q_2\right) < d(Q_1, Q_2)$. In other words, the existence of the d-support sequence satisfying (18) and (19) means that the composite operator of "repeatedly applying $B^\gamma$ for $|\mathcal{S}|$ times" (i.e. $(\mathcal{B}^\gamma)^{|\mathcal{S}|}$) guarantees to reduce the $L_\infty$-distance of every value function pair in $\mathcal{Q}$.

Moreover, note that the initial values of $Q_1$ and $Q_2$ do not determine how much percentage the distance between them will reduce – the values of $Q_1$ and $Q_2$ only affect what $a_{\max}$ is in (22), There are $|\mathcal{S}| \cdot |\mathcal{A}| \cdot |\mathcal{S}|$ possible transition probabilities in total, and $|\mathcal{S}|$ possible $\gamma$-values in (22), so no matter how $a_{\max}$ (which is a policy) and $\arg\max \mathbf{d}$ change over iterations, they just select a different subset from the $|\mathcal{S}|^3 \cdot |\mathcal{A}|$ possible terms. There are a finite number of such subsets, thus when the sum of the subset is less than 1, there must be an absolute upper bound $1 - r_{\min}$ for the subset sum. This upper bound ratio of distance reduction can be extremely close to 1, especially after repeating the process for $|\mathcal{S}|$ times, but still, it is a fixed number smaller than 1, and thus the operator $(\mathcal{B}^\gamma)^{|\mathcal{S}|}$ is a contraction mapping, and thus it has a unique fixed point (by the Banach fixed point theorem).

Our last step is to prove that the unique fixed point of the composite operator $(\mathcal{B}^\gamma)^{|\mathcal{S}|}$ is also the *unique* fixed (and limiting) point of the original Bellman operator $\mathcal{B}^\gamma$. It is easy to see that $\mathcal{B}^\gamma$ cannot have two fixed points, as otherwise their distance could not get reduced by repeatedly applying $\mathcal{B}^\gamma$ for $|\mathcal{S}|$ times. To show that $\mathcal{B}^\gamma$ does have *a* fixed point, we prove the following slightly stronger result (in the following lemma we use $\mathcal{B}$ to denote the generalized Bellman operator for brevity).

**Lemma 2.3.** *Under the condition of Theorem 1, let $Q^*$ be the unique fixed point of $(\mathcal{B})^{|\mathcal{S}|}$,*

$$Q^* = \lim_{n\to\infty} (\mathcal{B})^n Q, \quad \forall Q \in \mathcal{Q} \tag{24}$$

*and*

$$Q^* = \mathcal{B} Q^* \tag{25}$$

*Proof.* We first prove that (24) holds for all $Q^-$ with $Q^- \le \mathcal{B}Q^-$. [3] Because $\mathcal{B}$ is monotonic operator, for such $Q^-$ we have

$$Q^- \le \mathcal{B}Q^- \le (\mathcal{B})^2 Q^- \cdots \le (\mathcal{B})^{|S|} Q^- \cdots \le (\mathcal{B})^{2\cdot|S|} Q^- \cdots \le Q^* \tag{26}$$

where $Q^*$ being the limit of the above sequence is guaranteed by the contraction property of $\mathcal{B}^{|\mathcal{S}|}$. Recall that $Q_1 \le Q_2$ means $\forall s, a \;\; Q_1(s, a) \le Q_2(s, a)$ and that $d(Q_1, Q_2) = \max_{s,a} |Q_1(s,a) - Q_2(s,a)|$, we see that the value functions in the above (monotonic) sequence must have non-increasing distances to $Q^*$. On the other hand, as the subsequence $Q^-, (\mathcal{B})^{|\mathcal{S}|} Q^-, (\mathcal{B})^{2\cdot|\mathcal{S}|} Q^-, (\mathcal{B})^{3\cdot|\mathcal{S}|} Q^-, \ldots$ converges to $Q^*$, we know that for any $\epsilon > 0$, there exists an $i^*$ such that $d\left((\mathcal{B})^{i^*\cdot|\mathcal{S}|} Q^-, Q^*\right) < \epsilon$, and thus $d\left((\mathcal{B})^i Q^-, Q^*\right) < \epsilon$ for all integer $i > i^*$ due to the monotonicity, which literally means that the overall sequence (26) also converges to $Q^*$.

Now because $Q^* = \lim_{n\to\infty} (\mathcal{B})^n Q^-$, we must have $\mathcal{B}Q^* = \mathcal{B} \lim_{n\to\infty} (\mathcal{B})^n Q^- = \lim_{n\to\infty} (\mathcal{B})^n Q^- = Q^*$, thus (25) holds, and that $Q^*$ is a fixed point of $\mathcal{B}$.

---

[3] Such $Q^-$ guarantees to exist. In fact, every *on-policy value function* is such a $Q^-$; see A.3 for details.

Finally, because $Q^* = \mathcal{B}Q^*$, for any $Q \in \mathcal{Q}$ (not necessarily with $Q \leq \mathcal{B}Q$ this time), we have $d(\mathcal{B}Q, Q^*) = d(\mathcal{B}Q, \mathcal{B}Q^*) \geq d(Q, Q^*)$, where the inequality is by (17). Again, $d(\mathcal{B}Q, Q^*) \geq d(Q, Q^*)$ means the sequence (26) has non-increasing distances to $Q^*$, this time for all $Q \in \mathcal{Q}$. So, by the same logic as above, $Q^*$ must be the limit of the sequence $Q, \mathcal{B}Q, (\mathcal{B})^2 Q, \ldots$ – again, for all $Q \in \mathcal{Q}$ this time (not necessarily with $Q \leq \mathcal{B}Q$). $\qquad\square$

Note that the specific form of the $\gamma$-function does not play a role in the proof of Lemma 2.3 as presented above, so (25) and (24) apply to all $\mathcal{B}^\gamma$ that complies with the condition of Theorem 1. We thus have completed the proof of statement (1) and (2) in Theorem 1. $\qquad\square$

In comparison, the classic Bellman optimality property requires $\gamma(s) < 1$ at *every* state $s \in \mathcal{S}$, while Theorem 1 only requires $\gamma(s) < 1$ at terminal states. On the other hand, the classic result applies to all MDPs, while Theorem 1 is fundamentally based on unique structures of episodic learning process. Importantly, the episodic discounting function that we use in ELP – i.e. (3) – does satisfy the condition in Theorem 1.

### A.3    PROOF OF THEOREM 1 (3)

Finally, we prove statement (3) of Theorem 1 which asserts that $Q^*$, as the unique fixed point of the Bellman operator under episodic discounting, is indeed an optimal value function, in the sense that a $Q^*$-greedy policy is an optimal policy with respect to the undiscounted objective (1).

 (Theorem 1 (3)). *In any finite ELP, let $Q^*$ be the fixed point of $\mathcal{B}$ (i.e. the solution of (4)), let $\pi^*$ be a policy such that $\pi^*(a|s) > 0$ only if $Q^*(s, a) = \max_{\bar{a}} Q^*(s, \bar{a})$, then $J(\pi^*) = \max_{\pi \in \Pi} J(\pi)$, where $J$ is the episodic-reward objective* (1).

*Proof.* Every policy $\pi$ is coupled with a (unique) **on-policy value function** $Q_\pi$ which assigns values to $(s, a)$ pairs according to the conditional expectations of the episode-wise cumulative reward under the policy $\pi$. Formally,

$$Q_\pi(s, a) \doteq \mathop{\mathbf{E}}_{\{S_t, A_t\} \sim \pi} \left[ \sum_{t=1}^{T} R(S_t) \Big| S_0 = s, A_0 = a \right] \quad , \quad \forall (s, a) \in \mathcal{S} \times \mathcal{A} \qquad (27)$$

Comparing (27) with the definition of the episodic-reward objective $J$ (i.e. with (1)), and by the ELP conditions, we see that for any policy $\pi$, its performance score equals its on-policy *terminal values*:

$$J(\pi) = Q_\pi(s_\perp, a) \quad , \quad \forall s_\perp \in \mathcal{S}_\perp \,, \, \forall a \in \mathcal{A}. \qquad (28)$$

We will prove that for the $Q^*$-greedy policy $\pi^*$, we have $Q_{\pi^*} \geq Q_\pi$ for all $\pi \in \Pi$, which would entail that $J(\pi^*) \geq J(\pi)$.

First observe that $Q_{\pi^*} = Q^*$, that is, the value function $Q^*$ which induced the greedy policy $\pi^*$ is also the on-policy value function of $\pi^*$. Specifically, with the episodic $\gamma$-function (3), for any $(s, a)$, by the definition of $\pi^*$ we have

$$Q^*(s, a) = \mathcal{B}Q^*(s, a)$$

$$= \mathop{\mathbf{E}}_{S' \sim P(s,a)} \max_{a'} \left[ R(S') + \gamma(S') \cdot Q^*(S', a') \right]$$

$$= \mathop{\mathbf{E}}_{S' \sim P(s,a)} \mathop{\mathbf{E}}_{A' \sim \pi^*(S')} \left[ R(S') + \gamma(S') \cdot Q^*(S', A') \right] \qquad (29)$$

$$= \mathop{\mathbf{E}}_{\{S_1, A_1\} \sim \pi^*} \left[ R(S_1) + \gamma(S_1) \cdot Q^*(S_1, A_1) \Big| S_0 = s, A_0 = a \right]$$

$$= \mathop{\mathbf{E}}_{\{S_1, A_1, S_2, A_2\} \sim \pi^*} \left[ R(S_1) + \gamma(S_1) R(S_2) + \gamma(S_1)\gamma(S_2) \cdot Q^*(S_2, A_2) \,\Big|\, S_0 = s, A_0 = a \right]$$

$$= \mathop{\mathbf{E}}_{\{S_t, A_t\} \sim \pi^*} \left[ \sum_{t=1}^{T} R(S_t) \Big| S_0 = s, A_0 = a \right]$$

$$= Q_{\pi^*}(s, a) \qquad (30)$$

Connecting (29) and (30) and generalizing to arbitrary $\pi$ yields that for any policy $\pi \in \Pi$, we obtain $Q_\pi \leq \mathcal{B}Q_\pi$; specifically,

$$
\begin{aligned}
Q_\pi(s,a) &= \mathop{\mathbf{E}}_{S' \sim P(s,a)} \mathop{\mathbf{E}}_{A' \sim \pi(S')} \Big[ R(S') + \gamma(S') \cdot Q_\pi(S', A') \Big] \\
&\leq \mathop{\mathbf{E}}_{S' \sim P(s,a)} \max_{a'} \Big[ R(S') + \gamma(S') \cdot Q_\pi(S', a') \Big] \\
&= \mathcal{B}Q_\pi(s,a)
\end{aligned}
$$

In other words, the set $\{Q : \exists \pi \in \Pi, Q = Q_\pi\}$ is a subset of the set $\{Q : Q \leq \mathcal{B}Q\}$.

Now, because $Q^*$ is a maximum value-function of the set $\{Q \leq \mathcal{B}Q\}$ which contains all on-policy value functions as a subset, and because $Q^*$ itself is also an on-policy value function (as $Q^* = Q_{\pi^*}$), it must follow that $Q^*$ is also a maximum value-function of the subset $\{Q_\pi\}$. Thus we have proved that $Q_{\pi^*} = Q^* \geq Q_\pi$ for all $\pi \in \Pi$, as desired. $\qquad\square$

## B  PROOFS OF MINIMAX DUALITY

### B.1  PROOF OF LEMMA 2.1

(Lemma 2.1). *In any finite ELP, for any conjugate policy $\pi$, $Q^* \in \arg\min\limits_{Q \in \mathcal{Q}} \max\limits_{\boldsymbol{\lambda} \geq 0} \mathcal{L}_\pi(Q, \boldsymbol{\lambda})$.*

$Q^*$ is an optimal solution of (7) because $\mathcal{B}$ is monotonic, so for $Q$ with $Q \geq \mathcal{B}Q$ we have $Q \geq \mathcal{B}Q \geq (\mathcal{B})^2 Q \cdots \geq Q^*$. The objective function of (7) is some probabilistic average over the values at terminal states (and actions), which clearly attains minimum for the per-state-action minimum value function $Q^*$.

Due to standard Lagrangian duality, a value function is an optimal solution of (7) if and only if it is a minimax solution of (8). In particular, only value functions with $Q \geq \mathcal{B}Q$ can prevent $\max_{\boldsymbol{\lambda}} \mathcal{L}_\pi(Q, \boldsymbol{\lambda})$ from being arbitrarily large, and for those $Q$'s, $\max_{\boldsymbol{\lambda}} \mathcal{L}_\pi(Q, \boldsymbol{\lambda}) = \mathbf{E}_{\zeta \sim \pi} \Big[ Q(S_T, A_T) \Big]$ under the complementary slackness condition, whose value attains minimum at $Q^*$ as proved.

### B.2  PROOF OF LEMMA 2.2

(Lemma 2.2). *In any finite Episodic Learning Process $(\mathcal{S}, \mathcal{A}, P, R, \rho_0)$, for any conjugate policy $\pi$, let $\mathcal{L}_\pi$ be the corresponding Lagrangian, and let $\boldsymbol{\lambda}_\pi$ be the particular Lagrangian multipliers with $\boldsymbol{\lambda}_\pi(s,a) = \rho_\pi(s) \cdot \pi(a|s) \cdot \mathbf{E}_{\zeta \sim \pi}[T]$, where $\rho_\pi$ is the stationary distribution of $\pi$, then*

$$
\mathcal{L}_\pi(Q, \boldsymbol{\lambda}_\pi) = J(\pi) + \sum_{s \notin \mathcal{S}_\perp} \sum_{a \in \mathcal{A}} \boldsymbol{\lambda}_\pi(s,a) \cdot \Big( \max_{\bar{a}} Q(s, \bar{a}) - Q(s,a) \Big)
$$

By Proposition 8, with $f(s) = \mathbf{E}_{A \sim \pi(s)} \Big[ \mathbb{1}[s \in \mathcal{S}_\perp] \cdot Q(s, A) \Big]$, we have

$$
\mathop{\mathbf{E}}_{\zeta \sim \pi} \Big[ Q(S_T, A_T) \Big] = \mathop{\mathbf{E}}_{S_{1..T} \sim \pi} \Big[ \sum_{t=1}^{T} f(S_t) \Big] = \mathbf{E}_{\zeta \sim \pi}[T] \cdot \mathop{\mathbf{E}}_{S \sim \rho_\pi} \mathop{\mathbf{E}}_{A \sim \pi(S)} \Big[ \mathbb{1}[S \in \mathcal{S}_\perp] \cdot Q(S, A) \Big]
$$

So, for any $Q \in \mathcal{Q}$, we have

$$\mathcal{L}_\pi(Q, \boldsymbol{\lambda}_\pi)$$

$$=\mathbf{E}_{\zeta \sim \pi}[T] \cdot \mathop{\mathbf{E}}_{S,A \sim \rho_\pi} \left[ \mathbb{1}[S \in \mathcal{S}_\perp] \cdot Q(S,A) \right] + \mathbf{E}_{\zeta \sim \pi}[T] \cdot \mathop{\mathbf{E}}_{S,A \sim \rho_\pi} \left[ \mathcal{B}Q(S,A) - Q(S,A) \right]$$

$$=\mathbf{E}_{\zeta \sim \pi}[T] \cdot \mathop{\mathbf{E}}_{S,A \sim \rho_\pi} \left[ \mathbb{1}[S \in \mathcal{S}_\perp] \cdot Q(S,A) - Q(S,A) \right] +$$

$$\mathbf{E}_{\zeta \sim \pi}[T] \cdot \mathop{\mathbf{E}}_{S,A \sim \rho_\pi} \left[ \mathop{\mathbf{E}}_{S' \sim P(S,A)} \left[ R(S') + \gamma(S') \cdot \max_{a'} Q(S', a') \right] \right]$$

$$= - \mathbf{E}_{\zeta \sim \pi}[T] \cdot \mathop{\mathbf{E}}_{S,A \sim \rho_\pi} \left[ \mathbb{1}[S \notin \mathcal{S}_\perp] \cdot Q(S,A) \right] + \underline{\mathbf{E}_{\zeta \sim \pi}[T] \cdot \mathop{\mathbf{E}}_{S' \sim \rho_\pi} \left[ R(S') + \gamma(S') \cdot \max_{a'} Q(S', a') \right]}$$

$$=\underline{\mathbf{E}_{\zeta \sim \pi}[T] \cdot \mathop{\mathbf{E}}_{S \sim \rho_\pi} \left[ R(S) \right]} + \mathbf{E}_{\zeta \sim \pi}[T] \cdot \mathop{\mathbf{E}}_{S,A \sim \rho_\pi} \left[ \mathbb{1}[S \notin \mathcal{S}_\perp] \cdot \left( \max_a Q(S,a) - Q(S,A) \right) \right]$$

$$=\underline{J(\pi)} + \mathbf{E}_{\zeta \sim \pi}[T] \cdot \sum_{s \in \mathcal{S}, a \in \mathcal{A}} \rho_\pi(s) \cdot \pi(a|s) \cdot \mathbb{1}[s \notin \mathcal{S}_\perp] \cdot \left( \max_{\bar{a}} Q(s, \bar{a}) - Q(s,a) \right)$$

$$=J(\pi) + \mathbf{E}_{\zeta \sim \pi}[T] \cdot \sum_{s \in \mathcal{S} \setminus \mathcal{S}_\perp} \sum_{a \in \mathcal{A}} \rho_\pi(s) \cdot \pi(a|s) \cdot \left( \max_{\bar{a}} Q(s, \bar{a}) - Q(s,a) \right)$$

Note that in above for $\mathbf{E}_{\zeta \sim \pi}[T] \cdot \mathbf{E}_{S \sim \rho_\pi} \left[ R(S) \right] = J(\pi)$ we have again used the transformation of Proposition 8, with $f(s) = R(s)$.

## B.3 PROOF OF THEOREM 2

(ELP Minimax Theorem). *In any finite Episodic Learning Process* $(\mathcal{S}, \mathcal{A}, P, R, \rho_0)$, *if* $\mu \in \Pi$ *is an optimal policy, then its conjugate Lagrangian* $\mathcal{L}_\mu$ *has strong duality property, with*

$$\min_{Q \in \mathcal{Q}} \max_{\boldsymbol{\lambda} \geq 0} \mathcal{L}_\mu(Q, \boldsymbol{\lambda}) = \max_{\boldsymbol{\lambda} \geq 0} \min_{Q \in \mathcal{Q}} \mathcal{L}_\mu(Q, \boldsymbol{\lambda}) = J(\mu) \tag{31}$$

Let $\pi^*$ be a $Q^*$-greedy policy, which is thus an optimal policy.

For any conjugate policy $\pi$, since $Q^*$ is a minimax solution of $\mathcal{L}_\pi$ (Lemma 2.1), we have $\min_Q \max_{\boldsymbol{\lambda} \geq 0} \mathcal{L}_\pi(Q, \boldsymbol{\lambda}) = \max_{\boldsymbol{\lambda} \geq 0} \mathcal{L}_\pi(Q^*, \boldsymbol{\lambda}) = \mathbf{E}_\pi[Q^*(S_T, A_T)]$. By (30) in A.3 we have $\mathbf{E}_\pi[Q^*(S_T, \bar{A}_T)] = \mathbf{E}_\pi[Q_{\pi^*}(S_T, \bar{A}_T)]$. By (28) in A.3 we further have $\mathbf{E}_\pi[Q_{\pi^*}(S_T, A_T)] = J(\pi^*)$. Connecting these results together gives

$$\min_Q \max_{\boldsymbol{\lambda} \geq 0} \mathcal{L}_\pi(Q, \boldsymbol{\lambda}) = J(\pi^*). \tag{32}$$

Again for any conjugate policy $\pi$, due to Lemma 2.2, the Lagrangian function has the dual form (10) under the particular multiplier $\boldsymbol{\lambda}_\pi$, as copied below

$$\mathcal{L}_\pi(Q, \boldsymbol{\lambda}_\pi) = J(\pi) + \sum_{s \notin \mathcal{S}_\perp} \sum_{a \in \mathcal{A}} \boldsymbol{\lambda}_\pi(s, a) \cdot \left( \max_{\bar{a}} Q(s, \bar{a}) - Q(s,a) \right)$$

The first term above is a constant that does not change with $Q$, and in the second term both $\boldsymbol{\lambda}_\pi(s, a)$ and $\max_{\bar{a}} Q(s, \bar{a}) - Q(s,a)$ are nonnegative (for any $Q$ and $\pi$), so the sum of the two terms, i.e. $\mathcal{L}_\pi(Q, \boldsymbol{\lambda}_\pi)$, attains its minimum when $Q$ achieves complementary slackness with $\boldsymbol{\lambda}_\pi$, in which case

$$\min_{Q \in \mathcal{Q}} \mathcal{L}_\pi(Q, \boldsymbol{\lambda}_\pi) = J(\pi). \tag{33}$$

So now, when the conjugate policy $\pi$ is an optimal policy $\mu$, as assumed in the theorem, we have

$$\max_{\boldsymbol{\lambda} \geq 0} \min_{Q \in \mathcal{Q}} \mathcal{L}_\mu(Q, \boldsymbol{\lambda}) \geq \min_{Q \in \mathcal{Q}} \mathcal{L}_\mu(Q, \boldsymbol{\lambda}_\mu) = J(\mu) = J(\pi^*) = \min_Q \max_{\boldsymbol{\lambda} \geq 0} \mathcal{L}_\mu(Q, \boldsymbol{\lambda}).$$

Due to *weak minimax duality*, $\max_{\boldsymbol{\lambda} \geq 0} \min_{Q \in \mathcal{Q}} \mathcal{L}_\mu(Q, \boldsymbol{\lambda}) \leq \min_Q \max_{\boldsymbol{\lambda} \geq 0} \mathcal{L}_\mu(Q, \boldsymbol{\lambda})$, which universally holds, thus the inequality above must actually be an equality, as desired.

### B.4 PROOF OF PROPOSITION 3

(Proposition 3). *In any finite ELP, for any value function $Q \in \mathcal{Q}$ and any policy $\pi \in \Pi$, let $\lambda_\pi(s, a) = \rho_\pi(s, a) \cdot \mathbf{E}_{\zeta \sim \pi}[T]$, where $\rho_\pi(s, a) \doteq \rho_\pi(s) \cdot \pi(a|s)$, then*

$$\mathcal{L}_\pi(Q, \bar{\lambda}) \leq \mathcal{L}_\pi(Q, \lambda_\pi) \leq \mathcal{L}_\pi(\bar{Q}, \lambda_\pi) \quad , \quad \forall \bar{Q}, \bar{\lambda}$$

*if and only if*

*(1)* $\mathcal{B}Q(s, a) - Q(s, a) \leq 0 \qquad\qquad , \quad \forall(s, a) \in \mathcal{S} \times \mathcal{A}$

*(2)* $\rho_\pi(s, a) \cdot \Big( \mathcal{B}Q(s, a) - Q(s, a) \Big) = 0 \qquad , \quad \forall(s, a) \in \mathcal{S} \times \mathcal{A}$

*(3)* $\rho_\pi(s, a) \cdot \Big( \max_{\bar{a}} Q(s, \bar{a}) - Q(s, a) \Big) = 0 \quad , \quad \forall s \notin \mathcal{S}_\perp, a \in \mathcal{A}$

The "if" part is straightforward: Condition (1) and (2) immediately gives $\mathcal{L}_\pi(Q, \lambda_\pi) = \mathbf{E}_{\zeta \sim \pi}[Q(S_T, A_T)] = \max_{\bar{\lambda} \geq 0} \mathcal{L}_\pi(Q, \bar{\lambda})$. On the other hand, condition (3) entails that the second term in dual-form Lagrangian (10) is zero, so $\mathcal{L}_\pi(Q, \lambda_\pi) = J(\pi)$. By (33) in B.3 we have $\min_{\bar{Q}} \mathcal{L}_\pi(\bar{Q}, \lambda_\pi) = J(\pi)$, thus $\mathcal{L}_\pi(Q, \lambda_\pi) = \min_{\bar{Q}} \mathcal{L}_\pi(\bar{Q}, \lambda_\pi)$.

Now we prove the "only if" part, for which we resort to the general saddle-point condition: Under *given* conjugate policy $\pi$, a $(Q, \lambda)$ pair is a minimax equilibrium (or minimax saddle-point) of function $\mathcal{L}_\pi(Q, \lambda)$ if and only if

(i) $\min_{\bar{Q} \in \mathcal{Q}} \max_{\bar{\lambda} \geq 0} \mathcal{L}_\pi(\bar{Q}, \bar{\lambda}) = \max_{\bar{\lambda} \geq 0} \min_{\bar{Q} \in \mathcal{Q}} \mathcal{L}_\pi(\bar{Q}, \bar{\lambda}) = \mathcal{L}_\pi(Q, \lambda)$,

(ii) $Q \in \arg \min_{\bar{Q} \in \mathcal{Q}} \max_{\bar{\lambda} \geq 0} \mathcal{L}_\pi(\bar{Q}, \bar{\lambda})$,

(iii) $\lambda \in \arg \max_{\bar{\lambda} \geq 0} \min_{\bar{Q} \in \mathcal{Q}} \mathcal{L}_\pi(\bar{Q}, \bar{\lambda})$.

By condition (ii), if $(Q, \lambda_\pi)$ form a minimax equilibrium, then $Q$ must be a minimax value function. From B.1 we know that such minimax value function must have $Q \geq \mathcal{B}Q$ (condition (1)).

By condition (i) and by (32) in B.3 we know that $\mathcal{L}_\pi(Q, \lambda_\pi) = \min_{\bar{Q} \in \mathcal{Q}} \max_{\bar{\lambda} \geq 0} \mathcal{L}_\pi(\bar{Q}, \bar{\lambda}) = J(\pi^*) = \mathbf{E}_{\zeta \sim \pi}[Q(S_T, A_T)]$. In other words, $\sum_{s,a} \lambda_\pi(s, a) \cdot \big( \mathcal{B}Q(s, a) - Q(s, a) \big)$, as the second term in $\mathcal{L}_\pi(Q, \lambda_\pi)$, needs to be zero. Because $\lambda_\pi(s, a) = \rho_\pi(s, a) \cdot \mathbf{E}_{\zeta \sim \pi}[T]$ where $\mathbf{E}_{\zeta \sim \pi}[T] > 0$, and because $\mathcal{B}Q \leq Q$ as proved, the only way to make the term zero is to have $\rho_\pi(s, a) \cdot \big( \mathcal{B}Q(s, a) - Q(s, a) \big) = 0$ for each and every $(s, a)$ (condition (2)).

Moreover, since $\mathcal{L}_\pi(Q, \lambda_\pi) = \min_{\bar{Q}} \mathcal{L}_\pi(\bar{Q}, \lambda_\pi)$ as assumed, by (33) in B.3 we have $\mathcal{L}_\pi(Q, \lambda_\pi) = J(\pi)$, which means $\sum_{s \notin \mathcal{S}_\perp} \sum_{a \in \mathcal{A}} \rho_\pi(s, a) \cdot \mathbf{E}_{\zeta \sim \pi}[T] \cdot \big( \max_{\bar{a}} Q(s, \bar{a}) - Q(s, a) \big)$, as the second term of the dual-form Lagrangian (10), must be zero. Again, because $\mathbf{E}_{\zeta \sim \pi}[T] > 0$ and because $\max_{\bar{a}} Q(s, \bar{a}) - Q(s, a) \geq 0$ for all $(s, a)$, the only possibility is to have $\rho_\pi(s, a) \cdot \big( \max_{\bar{a}} Q(s, \bar{a}) - Q(s, a) \big) = 0$ for each $(s, a) \in \mathcal{S} \setminus \mathcal{S}_\perp \times \mathcal{A}$ (condition (3)).

## C MAXIMIN SADDLE POINTS VS MINIMAX SADDLE POINTS

### C.1 PROOF OF LEMMA 5.1

(Lemma 5.1). *In any finite ELP, $Q^*$ is an optimal solution of*

$$\max_Q \quad \mathbf{E}_{\zeta \sim \pi} \Big[ Q(S_T, A_T) \Big] \quad s.t. \quad Q(s, a) \leq \mathcal{B}Q(s, a) \ , \ \forall(s, a)$$

*for any conjugate policy $\pi \in \Pi$. Equivalently, $Q^* \in \arg \max_{Q \in \mathcal{Q}} \min_{\lambda \geq 0} \mathcal{L}_\pi(Q, \lambda)$.*

The proof is by symmetric argument with the one in B.1: Because $\mathcal{B}$ is monotonic, for $Q$ with $Q \leq \mathcal{B}Q$ we have $Q \leq \mathcal{B}Q \leq (\mathcal{B})^2 Q \cdots \leq Q^*$, thus $Q^*$ maximizes the objective $\mathbf{E}_{\zeta \sim \pi}[Q(S_T, A_T)]$ in a per-state-action manner.

## C.2 PROOF OF THEOREM 5

(Theorem 5). *In any finite ELP, for any conjugate policy $\pi$, let $Q_{\max}$ be an maximin value function of the Lagrangian $\mathcal{L}_\pi$ – i.e. let $Q_{\max}$ be an optimal solution of* (15) *– then $Q_{\max}$ is an optimal value function, in the sense that $Q_{\max}$-greedy policy maximizes the episodic-reward objective* (1).

As described in the proof idea, first observe that $Q_{\max}(s, a) \leq Q^*$ for all $(s, a) \in \mathcal{S} \times \mathcal{A}$ because $Q_{\max}$, as a feasible solution of (15), has $Q_{\max} \leq \mathcal{B}Q_{\max} \leq \mathcal{B}\mathcal{B}Q_{\max} \cdots \leq Q^*$.

Let $\mu$ be a $Q_{\max}$-greedy policy, so $\mu(a|s) > 0$ only if $Q_{\max}(s, a) = \max_{\bar{a}} Q_{\max}(s, \bar{a})$. We will prove that

$$\max_{\bar{a}} Q_{\max}(s, \bar{a}) = \max_{\bar{a}} Q^*(s, \bar{a}) \quad , \quad \forall s \in \mathcal{S}_\mu \setminus \mathcal{S}_\perp. \tag{34}$$

Note that (34) would necessarily imply that

$$\arg\max_a Q_{\max}(s, a) \subseteq \arg\max_a Q^*(s, a) \quad , \quad \forall s \in \mathcal{S}_\mu \setminus \mathcal{S}_\perp. \tag{35}$$

This is because $Q_{\max} \leq Q^*$ guarantees that for any action $a$ sub-optimal to $Q^*$, it can only have even lower value in $Q_{\max}$, so $Q_{\max}(s, a) \leq Q^*(s, a) < \max_{\bar{a}} Q^*(s, \bar{a}) = \max_{\bar{a}} Q_{\max}(s, \bar{a})$, in which $Q_{\max}(s, a) < \max_{\bar{a}} Q_{\max}(s, \bar{a})$ literally says that such an $a$ cannot be $Q_{\max}$-optimal either. (35) guarantees that $\mu$, a $Q_{\max}$-greedy policy, will only choose $Q^*$-optimal actions at every non-terminal state it may encounter since time $t \geq 1$. Such a $\mu$ is clearly an optimal policy in terms of episodic reward. Note that $\mu$'s choices on terminal states does not matter here as state-transitions in terminal steps are action-agnostic, due to the ELP condition.

Now, to prove (34), we first prove the following induction rule:

**Proposition 11.** *Under the context of Theorem 5, let $(s, a)$ be an arbitrary state-action pair, and let*

$$(\mathcal{S} \times \mathcal{A})_{next} \doteq \{(s', a') \in \mathcal{S} \times \mathcal{A} : s' \notin \mathcal{S}_\perp \ and \ P(s'|s, a) \cdot \mu(a'|s') > 0\}$$

*denote the set of all the non-terminal $(s', a')$ pairs that can directly follow $(s, a)$ under the $Q_{\max}$-greedy policy $\mu$, then*

$$Q_{\max}(s, a) = Q^*(s, a)$$
$$\Rightarrow Q_{\max}(s', a') = Q^*(s', a') \quad and \quad \max_{\bar{a}} Q_{\max}(s', \bar{a}) = \max_{\bar{a}} Q^*(s', \bar{a}) \ , \ \forall (s', a') \in (\mathcal{S} \times \mathcal{A})_{next}$$

*Proof.* Because $Q_{\max} \leq Q^*$ and $Q_{\max} \leq \mathcal{B}Q_{\max}$, we have

$$
\begin{aligned}
Q_{\max}(s, a) &\leq \mathcal{B}Q_{\max}(s, a) \\
&= \mathbf{E}_{S'}[R(S')] + \sum_{s' \in \mathcal{S}} P(s'|s, a) \cdot \gamma(s') \cdot \max_{\bar{a} \in \mathcal{A}} Q_{\max}(s', \bar{a}) \\
&\leq \mathbf{E}_{S'}[R(S')] + \sum_{s' \in \mathcal{S}} P(s'|s, a) \cdot \gamma(s') \cdot \max_{\bar{a} \in \mathcal{A}} Q^*(s', \bar{a}) \\
&= \mathcal{B}Q^*(s, a) \\
&= Q^*(s, a)
\end{aligned}
\tag{36}
$$

So, the premise $Q_{\max}(s, a) = Q^*(s, a)$ entails that the two inequalities in above must be equality, among which the second one – i.e. (36) – can be equality only if

$$\sum_{s' \in (\mathcal{S} \times \mathcal{A})_{next}} P(s'|s, a) \cdot \max_{\bar{a} \in \mathcal{A}} Q_{\max}(s', \bar{a}) = \sum_{s' \in (\mathcal{S} \times \mathcal{A})_{next}} P(s'|s, a) \cdot \max_{\bar{a} \in \mathcal{A}} Q^*(s', \bar{a}) \tag{37}$$

where $s' \in (\mathcal{S} \times \mathcal{A})_{next}$ is a slight abuse of notation which means $s'$ shows up in $(\mathcal{S} \times \mathcal{A})_{next}$ with some coupled $a'$ (or equivalently, $s' \notin \mathcal{S}_\perp$ and $P(s'|s, a) > 0$).

Now observe that for (37) to hold, the only possibility is that

$$\max_{\bar{a} \in \mathcal{A}} Q_{\max}(s', \bar{a}) = \max_{\bar{a} \in \mathcal{A}} Q^*(s', \bar{a}) \quad , \quad \forall s' \in (\mathcal{S} \times \mathcal{A})_{next} \tag{38}$$

as otherwise for those $s'$ on which (38) do not hold, it can only be $\max_{\bar{a} \in \mathcal{A}} Q_{\max}(s', \bar{a}) < \max_{\bar{a} \in \mathcal{A}} Q^*(s', \bar{a})$ (because $Q_{\max} \leq Q^*$); those $s'$ must all have positive weights in (37) (by

definition of $(\mathcal{S} \times \mathcal{A})_{\text{next}}$), and thus will cause a real loss at the LHS of (37) (and importantly, no other state in $(\mathcal{S} \times \mathcal{A})_{\text{next}}$ could claim a "gain" to compensate this loss, again because $Q_{\max} \leq Q^*$).

Next, to prove $Q_{\max}(s', a') = Q^*(s', a')$ for all $(s', a') \in (\mathcal{S} \times \mathcal{A})_{\text{next}}$ (given the premise), notice that for any of such $(s', a')$ we have

$$\max_{\bar{a} \in \mathcal{A}} Q_{\max}(s', \bar{a}) = Q_{\max}(s', a') \leq Q^*(s', a') \leq \max_{\bar{a} \in \mathcal{A}} Q^*(s', \bar{a}) \tag{39}$$

in which $\max_{\bar{a} \in \mathcal{A}} Q_{\max}(s', \bar{a}) = Q_{\max}(s', a')$ is because $a'$ is by definition a $Q_{\max}$-greedy action under $s'$, and $Q_{\max}(s', a') \leq Q^*(s', a')$ is (once again) because $Q_{\max} \leq Q^*$.

By (38) we know that the two ends of (39) actually equal to each other, so the inequalities in between must also be equality, and in particular $Q_{\max}(s', a') = Q^*(s', a')$. $\qquad \square$

Proposition 11 as proved above enables us to prove (34) by induction (which is enough to prove the whole theorem, as argued above). Specifically, because both $Q_{\max}$ and $Q^*$ are optimal solutions of (15), and because the objective in (15) is a distribution over only the terminal states (see B.1), it follows that $Q_{\max}$ and $Q^*$ must be equal on at least one terminal state $s_\perp$. Starting from this terminal state $s_\perp$ – as well as an arbitrary action $a_\perp$ under it – we have $Q_{\max}(s_\perp, a_\perp) = Q^*(s_\perp, a_\perp)$, thus by the induction rule of Proposition 11 we obtain $\max_{\bar{a}} Q_{\max}(s', \bar{a}) = \max_{\bar{a}} Q^*(s', \bar{a})$ and $Q_{\max}(s', a') = Q^*(s', a')$ for all $(s', a')$ in the $(\mathcal{S} \times \mathcal{A})_{\text{next}}$ set with respect to $(s, a) = (s_\perp, a_\perp)$; the latter enables us to expand the induction proof to all non-terminal states that are reachable by $\mu$.

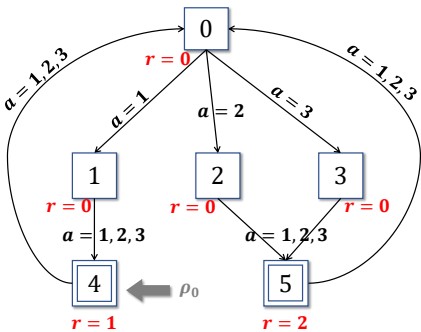

Figure 5: A copy of Figure 3

### C.3 AN COUNTER-EXAMPLE FOR THE $Q$-FORM MINIMAX VALUES IN ELPS

In this subsection we elaborate more about the counter-example as illustrated by Figure 3 in Section 5 (which is copied above). In this ELP, $\mathcal{S} = \{0, 1, 2, 3, 4, 5\}$, $\mathcal{A} = \{1, 2, 3\}$. State 4 and 5 are terminal states, from which any action leads to state 0. Choosing action $1, 2, 3$ under state 0 deterministically transits to state $1, 2, 3$, respectively. All actions under state 1 lead to state 4, and all actions under state 2 and 3 lead to state 5. The agent only receives non-zero rewards at terminal states, with $R(4) = 1$, $R(5) = 2$. The initial state is set to state 4 (i.e. $\rho_0(s) > 0$ only if $s = 4$).

The Bellman fixed-point $Q^*$ for this ELP is as follows:

- $Q^*(0, 1) = 1$, $Q^*(0, 2) = 2$, $Q^*(0, 3) = 2$
- $Q^*(1, a) = 1$, $\forall a$
- $Q^*(2, a) = Q^*(3, a) = 2$, $\forall a$
- $Q^*(4, a) = Q^*(5, a) = 2$, $\forall a$

An optimal policy of this ELP should only choose action 2 or 3, but not action 1, under state 0.

For minimax value functions, denoted by $Q_{\min}$, as they are optimal solutions of (7), we have

$$Q_{\min}(4, a) \;=\; 2 \;\geq\; \max \begin{cases} Q_{\min}(0, 1) & \geq & \max_a Q_{\min}(1, a) & \geq & 1 \\ Q_{\min}(0, 2) & \geq & \max_a Q_{\min}(2, a) & \geq & 2 \\ Q_{\min}(0, 3) & \geq & \max_a Q_{\min}(3, a) & \geq & 2 \end{cases}$$

The above minimax condition only imposes tight bounds for $Q^*$-optimal actions (e.g. action 2 and 3 under state 0), but leaves "flexibility" for actions sub-optimal to $Q^*$ (e.g. action 1 under state 0) as well as for *all* state-action pairs that follow an sub-optimal action (e.g. all actions under state 1). The consequence is that even constant value function $Q_{\min}(s,a) \equiv 2$ can be a minimax value function in this example, which is clearly sub-optimal, as discussed in Section 5.

In contrast, for maximin value functions, denoted by $Q_{\max}$, they are optimal solutions of (15), thus

$$Q_{\max}(4,a) \quad = \quad 2 \quad \leq \quad \max \begin{cases} Q_{\max}(0,1) & \leq & \max_a Q_{\max}(1,a) & \leq & 1 \\ Q_{\max}(0,2) & \leq & \max_a Q_{\max}(2,a) & \leq & 2 \\ Q_{\max}(0,3) & \leq & \max_a Q_{\max}(3,a) & \leq & 2 \end{cases}$$

We see that the maximin condition manages to imposes tight bounds for *at least one* $Q^*$-optimal action while at the same time can enforce *all* $Q^*$-sub-optimal actions to be still suboptimal to $Q_{\max}$. For example under state 0, $Q_{\max}(0,1)$ cannot exceed 1, while either $Q_{\max}(0,2)$ or $Q_{\max}(0,3)$ needs to be tight (i.e. $Q_{\max}(0,3) = 2$, or $Q_{\max}(0,2) = 2$, or both) so as to keep the maximum of the three no less than 2, as required.

Note that for both $Q_{\min}$ and $Q_{\max}$, the Lagrangian multiplier $\boldsymbol{\lambda}$ that forms equilibrium/saddle points with each of them (resp.) may not encode a policy, in general. In this example, for the constant $Q_{\min}$, we have $2 = Q_{\min}(0,1) = 0 + 1 \cdot Q_{\min}(1,a) > 1 + 0 \cdot Q_{\min}(4,a) = 1$, so $Q_{\min}(1,a) > \mathcal{B}Q_{\min}(1,a) = 1$ for all $a$, in which case its equilibrium multiplier $\boldsymbol{\lambda}$ has to have $\boldsymbol{\lambda}(1,a) = 0$ for all $a$, due to the equilibrium condition (2) proved in Proposition 3. Such an "all-zero" $\boldsymbol{\lambda}$ (on state 1) cannot be normalized into a policy.

Similarly, for the following maximum value function

- $Q_{\max}(4,a) = Q_{\max}(5,a) = 2$
- $Q_{\max}(0,1) = Q_{\max}(1,a) = 1$
- $Q_{\max}(0,2) = Q_{\max}(2,a) = 2$
- $Q_{\max}(0,3) = 1$
- $Q_{\max}(3,a) = 1.5$

we have $1 = Q_{\max}(0,3) < 0 + 1 \cdot Q_{\max}(3,a) < 2 + 0 \cdot Q_{\max}(5,a) = 2$, so $Q_{\max}(3,a) < \mathcal{B}Q_{\max}(3,a) = 2$ again for all $a$, so the multiplier corresponding to this $Q_{\max}$ still needs to be all zero at state 3 due to the complementary slackness condition, thus cannot be normalized at state 3.

### C.4 A COUNTER-EXAMPLE FOR THE $V$-FORM MINIMAX VALUES IN DISCOUNTED-MDPS

Moreover, the problems with minimax value functions as demonstrated above are not limited to $Q$-functions or to ELPs only, but seem to be fundamental issues rooted from the minimax structure. To see this, consider the *discounted-MDP* as shown in Figure 6 below.

To be strictly align with the related literature Dai et al. (2018); Cho and Wang (2017), the rewards are assigned to state-action pairs in this example. In this discounted-MDP, $\mathcal{S} = \{0,1,2,3\}$, $\mathcal{A} = \{1,2\}$. The initial state is set to state 0 (i.e. $\rho_0(s) = 1$ if $s = 0$, otherwise $\rho_0(s) = 0$). From state 0, taking

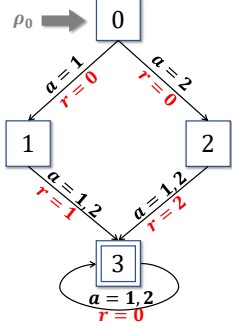

Figure 6: An example of discounted-MDP for studying saddle points of the $V$-form Lagrangian.

action $1, 2$ will deterministically goes to state $1, 2$, respectively, with zero reward obtained in this step. From state 1, any action leads to the absorbing state 3, with reward $R(1, a) = 1$ obtained (for all $a$). From state 2, any action leads to the same absorbing state 3, but with reward $R(2, a) = 2$ obtained (for all $a$). The absorbing state 3 will loop into itself forever, with zero reward obtained. An optimal policy in this discounted-MDP should choose action 2, not action 1, under state 0. The discounting constant $\gamma$ is set to 0.5, so the optimal discounted-reward performance is $V^*(0) = 1$.

The $V$-form Lagrangian of the discounted-MDP above is: Dai et al. (2018); Cho and Wang (2017)

$$(1 - \gamma) \cdot \mathop{\mathbf{E}}_{S_0 \sim \rho_0} \left[ V(S_0) \right] + \sum_{(s,a) \in \mathcal{S} \times \mathcal{A}} \lambda(s, a) \cdot \left( R(s, a) + \gamma \cdot \mathop{\mathbf{E}}_{S' \sim P(s,a)} \left[ V(S') \right] - V(s) \right) \quad (40)$$

A minimax state-value function $V_{\min}$ of the Lagrangian is an optimal solution of the Linear Programming problem (6), which has inspired some recently proposed RL algorithms (see Section 3). In this example, the LP can be more explicitly written as

$$\min_V \quad 0.5 \cdot V(0)$$
$$\text{s.t.} \quad V(0) \geq 0.5 \cdot V(1)$$
$$V(0) \geq 0.5 \cdot V(2)$$
$$V(1) \geq 0.5 \cdot V(3) + 1$$
$$V(2) \geq 0.5 \cdot V(3) + 2$$
$$V(3) \geq 0.5 \cdot V(3)$$

for which a possible optimal solution is: $V_{\min}(0) = 1$, $V_{\min}(1) = 2$, $V_{\min}(2) = 2$, $V_{\min}(3) = 0$, which assigns the same value to action 1 and 2 under state 0, thus is not an optimal value function.

## D  LAGRANGIAN MINIMIZATION FOR MACHINE TRANSLATION

### D.1  THE LAMIN1 ALGORITHM

As mentioned in Section 4, the idea of LAMIN1 is to minimize the smoothed Lagrangian $\mathcal{L}_\mu^\beta$ (with a small yet definite $\beta$) based on an unbiased gradient estimator of (13). For convenience, we copy (13) below:

$$\mathcal{L}_\mu^\beta(Q(\boldsymbol{w}), \boldsymbol{\lambda}_\mu) = \mathbf{E}_\mu[Q(S_T, A_T; \boldsymbol{w})] + \mathbf{E}_\mu[T] \cdot \mathop{\mathbf{E}}_{S,A,S' \sim \rho_\mu} \mathop{\mathbf{E}}_{A' \sim \pi_{Q(\boldsymbol{w})}^\beta(S')} \left[ \delta(S, A, S', A'; \boldsymbol{w}) \right]$$

where $\pi_{Q(\boldsymbol{w})}^\beta(a|s) \doteq \frac{exp\left(Q(s,a;\boldsymbol{w})/\beta\right)}{\sum_b exp\left(Q(s,b;\boldsymbol{w})/\beta\right)}$ is the Boltzmann distribution with temperature $\beta$, and $\delta(s, a, s', a'; \boldsymbol{w}) \doteq R(s') + \gamma_{\text{epi}}(s')Q(s', a'; \boldsymbol{w}) - Q(s, a; \boldsymbol{w})$ is the temporal-difference error. Algorithm 1 gives the psuedo-code of such an algorithm.

---

**Algorithm 1:** The LAMIN1 algorithm.

**Input:** A piece of rollout data made by an optimal policy, in the form of $\{s_t, a_t, r_t\}_{0,1,\ldots,n}$, where $t = T_1, T_2, \ldots, T_k$ are termination steps; A parameterized value function $Q(\boldsymbol{w})$ with initial weight vector $\boldsymbol{w}_0$; Learning rate $\alpha$; Boltzmann temperature $\beta$.

**for** *gradient update* $i = 0, 1, 2, \ldots$ **do**

$\quad \Delta \boldsymbol{w} \leftarrow \frac{1}{k} \sum_{t=\{T_1 \ldots T_k\}} \nabla_{\boldsymbol{w}} Q(s_t, a_t; \boldsymbol{w}) \Big|_{\boldsymbol{w} = \boldsymbol{w}_i} \; +$

$\quad \frac{n}{k} \cdot \frac{1}{n} \sum_{t=0}^{n-1} \gamma_{\text{epi}}(s_{t+1}) \nabla_{\boldsymbol{w}} \left( \sum_a \pi_{\boldsymbol{w}}^\beta(s_{t+1}, a) \cdot Q(s_{t+1}, a; \boldsymbol{w}) \right) - \nabla_{\boldsymbol{w}} Q(s_t, a_t; \boldsymbol{w}) \Big|_{\boldsymbol{w} = \boldsymbol{w}_i}$

$\quad \boldsymbol{w}_{i+1} \leftarrow \boldsymbol{w}_i - \alpha \cdot \Delta \boldsymbol{w}$

**Output:** A $Q(\boldsymbol{w})$-greedy policy.

---

Note that Algorithm 1 samples the stationary distribution $\rho_\mu$ in (13) by averaging over the rollout data of a bunch of episodes, which is unbiased thanks to the ergodicity of episodic learning Bojun

(2020). The overall LAMIN1 algorithm is thus a standard unbiased SGD procedure, which shares the generic convergence property of all SGD procedures (i.e. convergence to local minimum of (13) is guaranteed under properly annealed learning rate; see Goodfellow et al. (2016)).

As an implementation trick, the gradient computation in Algorithm 1 can utilize the following fact (for brevity and clarity, we omit the argument $s_{t+1}$ in $Q$ and $\pi$ in the following):

$$
\begin{aligned}
\nabla \pi_{\boldsymbol{w}}^{\beta}(a) &= \nabla \exp\Big(\log \frac{e^{Q(a;\boldsymbol{w})/\beta}}{\sum_b e^{Q(b;\boldsymbol{w})/\beta}}\Big) = \pi_{\boldsymbol{w}}^{\beta}(a) \cdot \nabla\Big(\log \frac{e^{Q(a;\boldsymbol{w})/\beta}}{\sum_b e^{Q(b;\boldsymbol{w})/\beta}}\Big) \\
&= \pi_{\boldsymbol{w}}^{\beta}(a) \cdot \Big(\nabla Q(a;\boldsymbol{w})/\beta \;-\; \nabla \log \sum_b e^{Q(b;\boldsymbol{w})/\beta}\Big) \\
&= \pi_{\boldsymbol{w}}^{\beta}(a) \cdot \Big(\nabla Q(a;\boldsymbol{w})/\beta \;-\; \frac{\sum_b e^{Q(b;\boldsymbol{w})/\beta} \cdot \nabla Q(b;\boldsymbol{w})/\beta}{\sum_c e^{Q(c;\boldsymbol{w})/\beta}}\Big) \\
&= \frac{1}{\beta} \cdot \Big(\pi_{\boldsymbol{w}}^{\beta}(a)\, \nabla Q(a;\boldsymbol{w}) - \pi_{\boldsymbol{w}}^{\beta}(a) \sum_b \pi_{\boldsymbol{w}}^{\beta}(b)\, \nabla Q(b;\boldsymbol{w})\Big)
\end{aligned}
$$

and so

$$
\begin{aligned}
&\nabla\Big(\sum_a \pi_{\boldsymbol{w}}^{\beta}(a) \cdot Q(a;\boldsymbol{w})\Big) \\
=\;& \sum_a \pi_{\boldsymbol{w}}^{\beta}(a) \cdot \nabla Q(a;\boldsymbol{w}) + \sum_a Q(a;\boldsymbol{w}) \cdot \nabla \pi_{\boldsymbol{w}}^{\beta}(a) \\
=\;& \sum_a \pi_{\boldsymbol{w}}^{\beta}(a) \cdot \nabla Q(a;\boldsymbol{w}) \;+\; \frac{1}{\beta} \cdot \sum_a \pi_{\boldsymbol{w}}^{\beta}(a)\, Q(a;\boldsymbol{w})\, \nabla Q(a;\boldsymbol{w}) \\
&- \frac{1}{\beta} \cdot \Big(\sum_a \pi_{\boldsymbol{w}}^{\beta}(a)\, Q(a;\boldsymbol{w})\Big) \cdot \Big(\sum_a \pi_{\boldsymbol{w}}^{\beta}(a)\, \nabla Q(a;\boldsymbol{w})\Big)
\end{aligned}
$$

Finally, we remark that the $\beta$-smoothing trick used in LAMIN1 is more than just an approximation heuristic, but may potentially play a role in correcting (to some extent) the sub-optimality bias of minimax value functions as discuss in Section 5. Specifically, the original Lagrangian function $\mathcal{L}_\mu(Q, \boldsymbol{\lambda}_\mu)$ fundamentally cannot distinguish optimal minimax-values from sub-optimal minimax-values, as the function attains its global minimum in both cases. In contrast, the smoothed Lagrangian $\mathcal{L}_\mu^{\beta}(Q, \boldsymbol{\lambda}_\mu)$ tends to reach lower value at optimal minimax-values than at sub-optimal minimax-values.

As an illustration, consider the special case where there is only one non-terminal state, under which the agent can only choose between two actions, 1 and 2, and suppose action 1 is the truly optimal. In this simple case, a value function can be represented by a tuple $(Q_1, Q_2)$, which indicating the value of action 1 and 2 respectively. A sub-optimal minimax-value function may have $Q_1 = Q_2$, as we showed in Section 5; in this case the smoothed Lagrangian $\mathcal{L}_\mu^{\beta=1}(Q, \boldsymbol{\lambda}_\mu)$ would still equal $J(\mu)$, which can be seen from the dual form of the smoothed Lagrangian:

$$
\mathcal{L}_\mu^{\beta=1}(Q, \boldsymbol{\lambda}_\mu) = J(\mu) + 1 \cdot \Big(\big(\frac{e^{Q_1} \cdot Q_1}{e^{Q_1} + e^{Q_2}} + \frac{e^{Q_2} \cdot Q_2}{e^{Q_1} + e^{Q_2}}\big) - Q_1\Big)
$$

On the other hand, an optimal minimax-value can make $\mathcal{L}_\mu^{\beta=1}(Q, \boldsymbol{\lambda}_\mu)$ lower than $J(\mu)$. For example, when $J(\mu) = Q_1 = 0$, we have $\mathcal{L}_\mu^{\beta=1}(Q, \boldsymbol{\lambda}_\mu) = \frac{e^{Q_2}}{1+e^{Q_2}} \cdot Q_2$, which attains $-0.28$ at $Q_2 = -1.3$. In other words, $\mathcal{L}_\mu^{\beta=1}(Q, \boldsymbol{\lambda}_\mu) < J(\mu) = 0$ under the optimal minimax-value $(0, -1.3)$, which is thus distinguished from the sub-optimal minimax-value $(0, 0)$ in smoothed Lagrangian minimization as LAMIN1 does.

## D.2 THE LAMIN2 ALGORITHM

The idea of LAMIN2 is to minimize the original Lagrangian function $\mathcal{L}_\mu$ based on the "local" gradient estimator (14). The consistency of the LAMIN2 estimator (with respect to $\nabla \mathcal{L}_\mu$) is characterized by Proposition 4, whose proof is given below.

(Proposition 4). *Let $\mathcal{A}$ be a finite action space, and let $Q(s, a; \boldsymbol{w})$ be a parameterized and differentiable value function that suggests a single best action $a_{\max}(\boldsymbol{w}^*) \doteq \arg\max_{a \in \mathcal{A}} Q(s, a; \boldsymbol{w}^*)$ when evaluating the actions under a given state $s$ with a given parameter vector $\boldsymbol{w}^*$, then*

$$\nabla_{\boldsymbol{w}} \max_{a \in \mathcal{A}} Q(s, a; \boldsymbol{w}) \Big|_{\boldsymbol{w}=\boldsymbol{w}^*} = \mathop{\mathbf{E}}_{a \sim \pi_{\max}(s; \boldsymbol{w}^*)} \Big[ \nabla_{\boldsymbol{w}} Q(s, a; \boldsymbol{w}) \Big] \Big|_{\boldsymbol{w}=\boldsymbol{w}^*} \tag{41}$$

*where $\pi_{\max}(\boldsymbol{w}^*)$ denote the $Q(\boldsymbol{w}^*)$-greedy policy.*

*Proof.* We will prove that for any component of $\boldsymbol{w}$,

$$\frac{\partial}{\partial w_i} \max_{a \in \mathcal{A}} Q(s, a; \boldsymbol{w}) \Big|_{\boldsymbol{w}=\boldsymbol{w}^*} = \frac{\partial}{\partial w_i} Q(s, a_{\max}(\boldsymbol{w}^*); \boldsymbol{w}) \Big|_{\boldsymbol{w}=\boldsymbol{w}^*} \tag{42}$$

which readily gives (41).

To prove (42), notice that

$$\frac{\partial}{\partial w_i} \max_{a \in \mathcal{A}} Q(s, a; \boldsymbol{w}) \Big|_{\boldsymbol{w}=\boldsymbol{w}^*} = \lim_{\Delta \to 0} \frac{Q(s, a_{\max}(\boldsymbol{w}^* + \Delta); \boldsymbol{w}^* + \Delta) - Q(s, a_{\max}(\boldsymbol{w}^*); \boldsymbol{w}^*)}{\Delta}$$

When there are a finite number of possible actions, there must be a non-zero gap between the best action and the second best action under $\boldsymbol{w}^*$. On the other hand, since $Q$ is differentiable, the change of action-values becomes infinitely small as $\Delta \to 0$, thus the order between the two best actions will remain the same for small enough $\Delta$, that is, $a_{\max}(\boldsymbol{w}^* + \Delta) = a_{\max}(\boldsymbol{w}^*)$ when $\Delta \to 0$. Therefore,

$$\lim_{\Delta \to 0} \frac{Q(s, a_{\max}(\boldsymbol{w}^* + \Delta); \boldsymbol{w}^* + \Delta) - Q(s, a_{\max}(\boldsymbol{w}^*); \boldsymbol{w}^*)}{\Delta}$$
$$= \lim_{\Delta \to 0} \frac{Q(s, a_{\max}(\boldsymbol{w}^*); \boldsymbol{w}^* + \Delta) - Q(s, a_{\max}(\boldsymbol{w}^*); \boldsymbol{w}^*)}{\Delta}$$
$$= \frac{\partial}{\partial w_i} Q(s, a_{\max}(\boldsymbol{w}^*); w) \Big|_{\boldsymbol{w}=\boldsymbol{w}^*}$$

$\square$

Note that when $Q(\boldsymbol{w})$ is a sophisticated real-valued function, such as a deep neural network with scalar output, the chance that two actions have *precisely* the same value under $Q(\boldsymbol{w})$ should be rare. Also, continuous action space can be densely quantized into a finite action space with *arbitrarily* small quantizing error, thus (41) should still approximately hold even for continuous action spaces.

---

**Algorithm 2:** The LAMIN2 algorithm.

---

**Input:** A piece of rollout data made by an optimal policy, in the form of $\{s_t, a_t, r_t\}_{0,1,\ldots,n}$, where $t = T_1, T_2, \ldots, T_k$ are termination steps; A parameterized value function $Q(\boldsymbol{w})$ with initial weight vector $\boldsymbol{w}_0$; Learning rate $\alpha$; Boltzmann temperature $\beta$.

**for** *gradient update* $i = 0, 1, 2, \ldots$ **do**

$\quad \Delta \boldsymbol{w} \leftarrow \frac{1}{k} \sum_{t=\{T_1 \ldots T_k\}} \nabla_{\boldsymbol{w}} Q(s_t, a_t; \boldsymbol{w}) \Big|_{\boldsymbol{w}=\boldsymbol{w}_i} +$

$\quad \quad \frac{n}{k} \cdot \frac{1}{n} \sum_{t=0}^{n-1} \gamma_{\text{epi}}(s_{t+1}) \left( \sum_a \pi_{\boldsymbol{w}_i}^{\beta}(s_{t+1}, a) \nabla_{\boldsymbol{w}} Q(s_{t+1}, a; \boldsymbol{w}) \right) - \nabla_{\boldsymbol{w}} Q(s_t, a_t; \boldsymbol{w}) \Big|_{\boldsymbol{w}=\boldsymbol{w}_i}$

$\quad \boldsymbol{w}_{i+1} \leftarrow \boldsymbol{w}_i - \alpha \cdot \Delta \boldsymbol{w}$

**Output:** A $Q(\boldsymbol{w})$-greedy policy.

---

Algorithm 2 gives the pseudo-code of LAMIN2 in a further generalized form, where the greedy policy $\pi_{\boldsymbol{w}_i}$ is replaced with the Boltzmann policy $\pi_{\boldsymbol{w}_i}^{\beta}$. As mentioned in Section 4, higher temperatures, such as $\beta = 1.0$, can slightly improve performance in our experiment results (for $0.5 - 0.8$ BLEU score).

### D.3 AN EPISODIC LEARNING FORMULATION OF MACHINE TRANSLATION

Many AI tasks are *sequence generation* problems, where we are given a context $X$, and are then asked to generate a sequence $Y = (\text{bos}, y_1 \ldots y_L, \text{eos})$ using *tokens* chosen from a given token space. Machine Translation (MT) is an example of such tasks, where the token space is the vocabulary of a target language, and the context $X$ is a sentence (or a sequence of sentences) in a source language. The choice of each token $y_t$ is conditioned on $X$ and on the partial output $Y_{<t} \doteq (\text{bos}, y_1 \ldots y_{t-1})$. In particular, $Y_{<1} = (\text{bos})$, and $Y_{<L+1} = (\text{bos}, y_1 \ldots y_L)$, conditioned on which the algorithm will generate the first token $y_1$ and the last token $y_{L+1} \doteq \text{eos}$, respectively.

The ELP formulation exactly captures the real-world MT tasks as described above. In the MT context, an episode is the translation of a given sentence. The first episode effectively starts with $S_1 = (X^{(1)}, \text{bos})$ where $X^{(1)}$ is a full source sentence. A learning agent then chooses a token $A_1 = y_1^{(1)} \in \Sigma_{\text{target}} \cup \{\text{eos}\}$, after which the environment state transits, deterministically, to $S_2 = (X^{(1)}, Y_{<2}^{(1)}) = (X^{(1)}, \text{bos}, y_1^{(1)})$. The agent keeps generating actions $A_t = y_t^{(1)}$ under each $S_t = (X^{(1)}, Y_{<t}^{(1)})$ until it outputs $A_{T-1} = \text{eos}$ at some step $T-1$, leading to terminal state $S_T = (X^{(1)}, Y^{(1)}) = (X^{(1)}, \text{bos}, y_1^{(1)} \ldots y_{T-2}^{(1)}, \text{eos})$. The agent will then make a normal action $A_T$ as in previous steps, which however makes no effect other than resetting the environment into $S_{T+1} = (X^{(2)}, \text{bos})$ from which the second translation episode begins. The process goes on episode after episode, generating a (theoretically infinite) sequence of translations $(X^{(1)}, Y^{(1)}), (X^{(2)}, Y^{(2)}), \ldots$, which collectively serve as the training data for the agent to learn better translation policies.

An episode of length $T$ as above results in a translation sentence $Y = (\text{bos}, y_1 \ldots y_L, \text{eos})$ which contains $L = T - 2$ "normal" tokens from $\Sigma_{\text{target}}$. As a common and necessary practice, most real-world MT systems impose a maximum translation length $H$ so that if an eos action did not show up after $H$ steps, the environment will transit to the terminal state $S_{H+2} = (X, \text{bos}, y_1 \ldots y_H, \text{eos})$ even if the agent continues to output normal token $A_{H+1} \in \Sigma_{\text{target}}$ at step $H+1$. When maximum translation length is applied, the corresponding episodic learning model of MT has bounded episode length, thus satisfies the ELP Condition (3) above. Such a formulation considers the mechanism of maximum translation length as a fundamental part of *learning-based* MT task specification that is essential in helping *learning* agents (which may not master when to output eos) to escape from long and meaningless translation episodes which may otherwise lead to ill-conditioned training data.

To comply with ELP Condition (2), we prescribe $\rho_0$ to be an arbitrary distribution over the set of terminal states, i.e. over all source-target sentence pairs $(X, Y)$ where $Y$ is complete sentence ending with eos. As with other terminal steps, no matter what $S_0$ and $A_0$ are, the next state $S_1$ will follow the same distribution, denoted as $\rho_1$, which specifies the distribution of the source sentences that the agent will receive in *each* episode (ELP Condition (1)).

Finally, the agent receives a scalar reward $R(X, Y) \in [0, 100]$ at each terminal state, based on the translation quality of $Y$ (with respect to $X$). The reward is zero at all non-terminal states (which has only partial translations). With this reward function, the expected episodic-reward objective $J(\pi)$ of a translation policy $\pi$ corresponds exactly to its average scores under the chosen translation metric.

The ELP formulation of MT as discussed above can be formally summarized as follows:

- $\mathcal{S} = (\Sigma_{\text{source}})^H \times \{\text{bos}\} \times (\Sigma_{\text{target}})^H \times \{\text{␣}, \text{eos}\}$

- $\mathcal{A} = \Sigma_{\text{target}} \cup \{\text{eos}\}$

- $R(s) = \begin{cases} \texttt{metric}\big(X(s), Y(s)\big) & s \in \mathcal{S}_\perp \\ 0 & s \notin \mathcal{S}_\perp \end{cases}$ , where $\mathcal{S}_\perp = \{(X, Y) : Y \text{ ends with } \text{eos}\}$

- $P(s'|s, a) = \begin{cases} \rho_1(s') & \text{if } s \in \mathcal{S}_\perp \\ \mathbb{1}[\, s' = (s, a)\,] & \text{if } s \notin \mathcal{S}_\perp \text{ and } |Y(s)| < H \\ \mathbb{1}[\, s' = (s, \text{eos})\,] & \text{if } s \notin \mathcal{S}_\perp \text{ and } |Y(s)| = H \end{cases}$

- $\rho_0(s) > 0$ only if $s \in \mathcal{S}_\perp$

Note that in above both $\mathcal{S}$ and $\mathcal{A}$ are finite sets, in which case the model is a *finite* episodic learning process. For real-world machine translation, we typically have $|\mathcal{S}| < 40000^{2048} \times 40000^{2048} \times 2$ and $|\mathcal{A}| < 40000 + 1$.

The experimentation code in Supplementary Material contains a faithful implementation in Python of the formulation presented here.

## D.4 EXPERIMENTATION DETAILS

We tested our algorithmic idea using the WMT'14 NewsTest English→German (en2de) dataset [4]. The data was pre-processed and post-processed using the BPE tokenizer provided by YouToken-ToMe [5], with shared vocabulary of size 37000. We used SacreBLEU Post (2018) as the translation metric and the standard TransformerBase neural network Vaswani et al. (2017), which is known to achieve a BLEU score of 27.3 on the WMT'14 dataset under the state-of-the-art method of MLE-based learning Vaswani et al. (2017).

We trained the model on the same 4.5 millions sentence pairs in the WMT'14 data set for $100,000$ gradient updates on a V100 GPU, with the same mini-batch size (and token-padding strategy) and learning rate schedule as recommended by Vaswani et al. (2017). A dropout rate of $0.1$ is applied. The learned model is then used as search heuristic in the *vanilla-beam-search decoding* (e.g. see Algorithm 1 in Stahlberg and Byrne (2019)), with a beam size of $4$. Empirically, we found that some more performance gain can be obtained by adding more tricks, such as modestly increasing the beam size (say, to 10), adding length penalty factor in the search heuristic (see Wu et al. (2016)), and model averaging (see Vaswani et al. (2017)), but we chose to exclude these tricks in our performance report so as to keep our algorithm simple and easy to implement. Our experimentation code and scripts can be found in Supplementary Material for reproduction.

The following tables give the numerical values of the performance scores shown in Figure 1.

| $\beta$ | BLEU@100k |
|---|---|
| 0.01 | 27.4 |
| 0.25 | 27.0 |
| 0.5 | 26.7 |
| 0.75 | 26.6 |
| 1.0 | 26.6 |

Table 1: Corpus BLEU scores of LAMIN1 (i.e. Algorithm 1) under different temperature $\beta$.

| $\beta$ | BLEU@100k |
|---|---|
| 0.01 | 26.0 |
| 0.2 | 26.2 |
| 0.4 | 26.8 |
| 0.6 | 26.2 |
| 0.8 | 26.3 |
| 1.0 | 26.8 |

Table 2: Corpus BLEU scores of LAMIN2 (i.e. Algorithm 2) under different temperature $\beta$.

---

[4]https://nlp.stanford.edu/projects/nmt/
[5]https://github.com/VKCOM/YouTokenToMe

