# OpenReview forum: "Lagrangian Method for Episodic Learning"
_ICLR.cc/2022/Conference — ICLR 2022 Submitted_

### Official Review · Reviewer_fdL5 · 2021-11-03

**Correctness:** 3
**Technical Novelty And Significance:** 3
**Empirical Novelty And Significance:** 2
**Recommendation:** 6
**Confidence:** 3

**Main Review:**

**strengths**

The theory of episodic RL recently attracted a lot of research interests in the ML community.  This paper complements the theory of episodic RL with interesting new results with good insight.

The overall presentation of this paper is well organized and follows a concentrated storyline. The authors also conducted well-designed machine translation experiments to demonstrate the usefulness of the derived learning algorithms and provide good insight connecting experiment observation and optimization theory.



**weaknesses**

(1) The major weakness of this paper is:  It has an implicit assumption/restriction that the reward function of the problem must solely depend on the current state.  In most MDP/RL problems, the reward function should also depend on the action (and possibly the next state.)  This weakness tremendously restricts the usefulness of this paper.

(2) As a paper intending to establish new theories, it occasionally missed important points in some key steps.  Below are a few examples:

(2.1) The constrained optimization reformulation (7) from (4) is abrupt.  (4) is for general MDP while (7) is specifically for episodic learning. I expected the authors to explain how T comes into play.

(2.2) The complementary slackness argument above (30) seems problematic.  The complementary slackness holds only under optimal (primal, dual) pairs.  Ignoring the MDP context, consider min f(x) s.t., g(x)<=0.  For a general given \lambda,  min f(x) + \lambda g(x) is not necessarily attained with a x such that g(x) = 0.   In the context of (30) to prove strong duality, it does not seem we are assuming any property of the dual variables.

**Summary Of The Paper:**

This paper studies the Q learning in episodic learning from a Lagrangian formulation of the Q-form Bellman optimality equation.   On the theory side, this paper studies the (1) fixed point of the bellman optimality operator when the discounting factor for non-terminal states are 1; (2) strong duality of the considered constrained optimization, i.e., saddle point,  reformulation; and (3)maximin type saddle points.  On the empirical side, this paper studies the efficacy of the minimax imitation learning algorithm for a machine translation application.

**Summary Of The Review:**

I recommend "6: marginally above the acceptance threshold" by assuming the technical weakness (2) is not flaws of the paper but rather imperfectness of presentation.  However, I strongly suggest the authors conduct certain clarifications in the final version if this paper is accepted.

---

> ### Author Response · Authors · 2021-11-09
> **Our state-based reward formulation is not a restriction**
>
> This response is dedicated to weakness (1) as mentioned in the review. We believe there is a substantial misunderstanding here: **our state-based reward formulation is not a restriction/assumption in any sense**.
>
> (1) Let us first show that our state-based reward formulation actually subsumes action-based reward formulations. Specifically, for any MDP with state space $S$ and "reward+state" transition function $p(s',r'|s,a)$ -- we believe this is the most general action-based reward formulation which subsumes all other action-based-reward variants (see Sutton&Barto 2018, Chapter 3.1)  -- it is always precisely equivalent to a MDP with "augmented" state space $Z = S \times \mathbb{R}$, state-transition function $\mathcal{P}(z'|z,a)$, and state-based reward function $R(z)$, where for any "augmented state" $z \triangleq (s,r)$,
> $$R(z)=R\big(~ (s,r) ~\big) \triangleq r$$
> and
> $$\mathcal{P}(z'|z,a)=\mathcal{P}\big((s',r')|(s,r),a\big) \triangleq p(s',r'|s,a).$$
> The idea above is that we can always think of a real-valued reward as just one (augmented) dimension of the state, and the state-based reward function $R(z)$ simply reads out the value of this dimension from a given state. Note that in such a formulation, $R(z)$ does **not** account for what have "caused" this reward; instead, $\mathcal{P}$ does (and in our formulation a reward is still "caused" by the action in the last step).
>
> (2) Our MDP formulation of machine translation, as formally presented in Appendix D.2, is a demonstration of the state-based reward formulation. Note that every reward in the translation-MDP is dependent to the action in the last step (specifically, we only observe non-zero reward if the last action is EOS). We also invite the reviewer to check our python code in supplementary material, specifically the brl/core/env.py file, in which we implemented a GymEnv class which translates any openai-gym environment into a state-based-reward environment via exactly the state-augmentation trick.
>
> (3) Note that in action-based reward formulations, the reward term (e.g. in the Bellman equation) is usually the reward of the **current** action (and state), while in state-based reward formulation, it is the the reward of the **next** state (e.g. see Eq.(4) in our paper). This detail may help explain why missing the explicit action dependency still lead to equivalent formulation (because the impact of the action takes effect "through" its impact on the next state).
>
> We believe the above three points are already sufficient to prove our central thesis (that our state-based reward formulation is not a restriction/assumption), but let us give some additional comments on this topic:
>
> (4) We are not the first authors who've used state-based reward formulation. For example, [Schulman et al, ICML'15], which proposed the classic TRPO algorithm and is cited 4000+ times now, used the same state-based reward formulation as ours. As another example, [Bojun, NeurIPS'20], which proposed the ELP formulation that our paper is based on, also used state-based reward formulation, and has discussed this formulation choice of state-based reward in its Appendix A.
>
> (5) If action-based and state-based reward are equivalent to each other, why did we choose to use the latter? For this particular paper, the reasons are three fold:
>
> (5.1) Our paper is based on the ELP framework as formalized in [Bojun, NeurIPS'20], which used state-based reward as mentioned.
>
> (5.2) We found that state-based reward indeed brought mathematical convenience to the theoretical analysis in our specific paper; for example, with state-based reward the objective function $J(\pi)$ can be written as an state-wise averaged reward, which is a crucial step in our proof of Lemma 2.2 (and this lemma in turn is crucial to our proof of the strong duality property in Theorem 2).
>
> (5.3) and this is perhaps the strongest motivation actually -- We think state-based reward formulation better reflects the **physical reality** of how the reward signal is actually extracted from the real-world environment. In our machine translation example, the reward (i.e. the translation quality) is indeed calculated from a *state* (i.e. a sentence pair). In Atari games, the reward (i.e. the change of game score) is indeed calculated from the game *state* resulted from an action like move-up (instead of directly from the move-up action). Even in robotics applications where an action appears to directly lead to an energy loss, the true amount of energy loss in the real world is still collected by reading the *state* of a meter right after the action.
>
> As the reviewer considers our reward formulation as "the major weakness of this paper" that "tremendously restricts the usefulness of this paper", we hope the above arguments has completely resolved the reviewer's concern on this issue, and hope it helps the reviewer to re-evaluate the value of this paper.

---

> ### Author Response · Authors · 2021-11-09
> **Regarding Eq.(4) and Eq.(7)**
>
> This response is dedicated to weakness (2.1) as mentioned in the review.
>
> "*(4) is for general MDP*"
>
> No, Eq.(4) is for episodic learning only. In particular, notice the $\gamma_{epi}$ term in (4), which is defined (in Eq.(3)) specifically for the episodic learning scenario. Please also see the paragraph right above (4), which explicitly explained this in more details. Both (4) and (7) are for the same problem setting (i.e. episodic learning), and (7) is a variational reformulation of (4).
>
>  "*I expected the authors to explain how T comes into play*"
>
> Thanks for the question. The terminal states $S_T$ (and terminal actions $A_T$) in (7) is in analogy to the initial states (which follows $\rho_0$) in (6). The involvement of both initial states as in the traditional V-form problem (6) and of the terminal states as in the Q-form problem (7) is due to the model-free nature of learning:
>
> In original dynamic programming literature (e.g. [Puterman, 1994]), the objective function of the (V-form) variational formulation was an average/sum over **all** states, for example in the form of $\sum_{s\in S} V(s)$. But such "whole-space" average/sum is not realistic in learning settings where the agent cannot easily sweep the state space (or cannot even sample the state space with uniform distribution). So, when the same variational idea is adapted to learning settings, such as in this paper,  the objective function of the variational problem (or equivalently, the first term of the Lagrangian function) needs to be something that the agent is able to estimate during the learning.
>
> In previous works (e.g. see the references in the paragraph right below (6)), the variational objective was typically chosen to be the average over initial states, i.e. $\sum_{s} \rho_0(s) V(s)$, which can be estimated in model-free rollouts.
>
> In our work, we replace the initial states with the "final states" (i.e. terminal states), whose average can be similarly estimated in model-free rollouts. But different from initial states whose distribution is fixed under any policy, the distribution of terminal states actually depend on the behavior policy, which brought in the concept of "conjugate policy" in our Q-form Lagrangian function. It turns out that such a mathematical construction can lead to the strong duality property as proved in our paper. Please see the proof of Lemma 2.2 for how precisely the terminal states/actions as well as the conjugate policy have played a role in bridging the logic of the proof.
>
> Hope the above provides some intuition behind the T in (7). In the revised version, we will try to add a few sentences in the paragraph right after (7) to discuss more about the involvement of T here, if the reviewer thinks the above explanation is helpful.

---

> ### Author Response · Authors · 2021-11-09
> **More elaboration on the complementary slackness argument**
>
> This response is dedicated to weakness (2.2) as mentioned in the review.
>
> We agree that, as a complete proof, the complementary slackness argument for the $Q$ and $\lambda$ in Eq.(10) indeed deserves some more elaborations. Yet, we believe our argument here is correct, and let us explain below.
>
> "*For a general given \lambda, min f(x) + \lambda g(x)*"
>
> This analogy is crucially different from what we are considering in that complementary slackness argument in that, in our argument we were talking about the complementary slackness for (10) which is the **dual form** of the original Lagrangian function. For convenience of discussion, we copy Eq.(10) below as follows:
> $$
> L_{\pi}(Q,\lambda_\pi) = A + B
> $$
> where
> $$
> A = J(\pi)
> $$
> and
> $$
> B = \sum_{s,a} \lambda_\pi (s,a) \cdot \Big(  \max_{\bar{a}} Q(s,\bar{a}) - Q(s,a) \Big)
> $$
> Please note that the first term $A$  -- i.e. $J(\pi)$ -- is a **constant** whose value does not change with different $Q$. Consequently, for a $Q$ such that $Q(s,a)=\max_{\bar{a}} Q(s,\bar{a})$ for all the $(s,a)$ pairs where $\lambda_\pi (s,a)>0$ -- such a $Q$ certainly exists for any **given** $\lambda_\pi$ -- the second term $B$ will be $0$ in which case $L_{\pi}(Q,\lambda_\pi) = A=J(\pi)$. On the other hand, $0$ is the smallest value that $B$ can attain because both $\lambda_\pi (s,a)$ and $\max_{\bar{a}} Q(s,\bar{a}) - Q(s,a)$ are non-negative. Therefore, $L_{\pi}(Q,\lambda_\pi)$ can attain $J(\pi)$ but not lower, thus the Eq.(30).
>
> So, in summary, we agree with the reviewer that for a **general** given \lambda, min f(x) + \lambda g(x) does not necessarily attain minimum with g(x)=0 but the key insight here is that we are not dealing with a general constrained optimization -- the **special** structure proved by Lemma 2.2 enables the **special** dual-form expression of the Lagrangian in our problem, in which the first term is not f(x) but something independent to x.
>
> We will add this elaboration to the paragraph above (30) if the reviewer finds it helpful to clarify doubts.

---

### Official Review · Reviewer_DWfZ · 2021-11-03

**Correctness:** 3
**Technical Novelty And Significance:** 2
**Empirical Novelty And Significance:** 2
**Recommendation:** 5
**Confidence:** 3

**Main Review:**

The theoretical results to my knowledge is new and would cover some applications. The quality of the presentation of this paper is not good. It is hard to find a clear roadmap to see what are the major contributions. My main concerns about this work are as follows:

1) What are the challenges of showing the strong duality from the linear to nonlinear case.

2) Why does this work focus on the episodic leaning process? Will the strong duality results be applicable to other cases? What are the unique features of ELP so that the strong duality holds?

3) The organization of this paper is quite wired. Why section 5 is before the numerical section? and there is no simulation result to justify the theorem shown in section 5.

4) There is no any convergence analysis of the proposed imitation learning algorithm.

5) Besides algorithm1, is there a general algorithmic framework based on this nonlinear.

6) Numerical results are limited. The ELP example shown in Figure 2 is rather artificial.

**Summary Of The Paper:**

In this work, the authors consider nonlinear Q-form Lagrangian function and show corresponding strong duality property. The main contributions are 1) the new proof of showing the duality gap as zero from a minmax perspective; 2) the generality of the theory with applications to machine translation tasks. A imitation learning algorithm is proposed as well and it turns out that it works well compared with the existing one.

**Summary Of The Review:**

In summary, this work indeed showed some progress regarding the nonlinear Q-function evaluation, but the quality of presenting the significance and challenges of having these results is not enough to be accepted at this stage.

---

> ### Author Response · Authors · 2021-11-10
> **Regarding roadmap of contributions**
>
> This response is dedicated to the reviewer's confusion about the roadmap of our contributions.
>
> "*It is hard to find a clear roadmap to see what are the major contributions.*"
>
> It's sad to hear that the reviewer felt confused about this because we actually spent four paragraphs in the Introduction section (from the sentence "*this paper provides a new perspective to the foundation of this elegant idea through the following contributions:*" down to the end of the section) exactly for giving a half-page summary of such a roadmap of contributions. Each paragraph summarized one contribution of our paper (so there are four major contributions in total), and each with a section number linked as the pointer. **It would be helpful for us to address the reviewer's concern if the reviewer could point out specifically which part of these paragraphs is unclear**.
>
> Among others, we feel the reviewer's current summary of our paper may have missed some important aspects of contributions of our paper. Specifically,
>
> -- Regarding the strong duality property, our paper did not only give "a new proof"; what we've proved is a new phenomenon. Specifically, strong duality property for the linear V-form Lagrangian is well-proved and our paper is not giving another proof of it. Instead, our paper proved strong duality for the nonlinear Q-form Lagrangian, for which the conclusion itself (not its proof) is a new discovery to our best knowledge.
>
> --  A major contribution of our paper is our discussion on the maximin-type saddle points of the Lagrangian, as well as on the limit of minimax-type saddle points on which the previous literature has almost exclusively focused. We believe this part of our paper pointed to a completely new direction of research.
>
> -- Another major contribution of our paper is to completely shift to the episodic-discounting framework throughout the paper, for both theoretical and empirical parts. To large extent, the "now-standard" discounted-MDP framework was historically introduced to bypass some technical difficulties, such as to directly enforce the contraction of the Bellman operator. While some economic/psychological applications do match the discounted-MDP setting, we believe that *much more* current AI practices deal with episodic learning tasks instead (e.g. atari games, locomotion, route navigation, Go/chess, translation, dialogue system, image captioning, etc). Deprecating the discounted-MDP setting has been an active movement; for example, (Sutton&Barto, 2018) dedicated its Section 10.4 on this issue, and the DP community has subsumed the discounted-MDP setting as a special case of an finite-termination setting for long time (e.g. see "Neuro-Dynamic Programming" by Bertsekas). Our paper contributes to this movement, closing a gap between theory and practice.

---

> ### Author Response · Authors · 2021-11-10
> **Regarding our presentation on significance and challenge**
>
> This response is dedicated to the reviewer's concern about our presentation on significance and challenge, which seems to be a main concern of the reviewer.
>
> (1) "*What are the challenges of showing the strong duality from the linear to nonlinear case?*"
>
> The two paragraphs right after we introduced the linear case (i.e. right below Eq.(6)) discussed exactly this. Basically, the challenge is that you can't use a "generic solution/methodology" to deal with the nonlinear case any more. There is no standard LP duality any more, so strong duality property does not hold "automatically" in nonlinear case (which can be taken for granted in linear case) and there is no generic method to study the duality property in nonlinear case to our knowledge. Also you can't readily utilize the many standard tricks in linear algebra to manipulate your nonlinear objects.
>
> (2) "*Why does this work focus on the episodic leaning process? Will the strong duality results be applicable to other cases? What are the unique features of ELP so that the strong duality holds?*"
>
> As we mentioned in another response to the reviewer, we focused on episodic learning process because of its high real-world correspondence. Our paper has explicitly pointed this out, e.g., in section 1 we wrote "*previous  works  are  mostly  limited  to  the  discounted-reward  setting,  which  bypass  some technical difficulties Nachum and Dai (2020), but deviate from the practical settings of many (if not most) real-world tasks.*", and in section 2 we wrote "*Following White (2017) and Bojun (2020), we use the ELP formalism to study finite-time decision tasks, which arguably account for most AI tasks encountered in current practice.*"
>
> In particular, we stress that we focus on episodic learning **not** because we think ELP is mathematically unique to our results or that our results do not apply to other cases (and we never tried to hint on that in the paper). It could be an interesting future work to extend our results to other MDP settings, but again, we chose to work on episodic learning because of its practical relevance and importance (at least for the tasks that we are looking at, such as atari games, route navigation, Go/chess, goal-oriented locomotion, translation, dialogue system, image captioning, etc).
>
> (3) "*The organization of this paper is quite wired. Why section 5 is before the numerical section? and there is no simulation result to justify the theorem shown in section 5.*"
>
> Section 5 (which is on the maximin-type saddle points) is after Section 4 (which is on the minimax-type saddle points) because Section 4 logically follows Section 3 (which is also on minimax-type saddle points) by turning the theory developed in Section 3 into practical algorithms and empirical results.
>
> We guess the reviewer might be expecting the popular "theory -> algorithm/pseudo-code -> experiment -> discussion" structure, thus might get confused to see a theory section after an experiment section as in our paper. But in our humble opinion, we don't think a paper *has to* be divided by methodology (theoretical vs empirical). We believe it's also proper to organize a paper according to which sub-topics the sections are talking about. In our case, Section 2 sets the stage for later sections, Section 3&4 focus on the usually-considered primal-form variational treatment of Bellman equation (which leads to minimax-type saddle points of the Lagrangian), and Section 5 focuses on the rarely-considered dual-form variational treatment (which leads to maximin-type saddle points). We feel this arrangement better narrates a storyline of our findings.
>
> As a side remark, note that we indeed organized the appendix sections in the theory -> experiment order (in particular, appendix C which supports section 5 is before appendix D which supports section 4). This is because readers are not expected to read appendix sections sequentially but are more likely to reach each appendix section separately following the pointers in the main text, thus maintaining clear storyline (as we've tried to do in the main text) is considered less important in the appendix.
>
> Regarding "simulation results to justify the theorem in Section 5", there is only one theorem in Section 5, that is Theorem 4, which has been rigorously proved without any restrictive assumption in Appendix C.2 (with proof idea given in Section 5). We are a bit not sure about this question actually, but we think a mathematically proved theorem does not need to be further "justified" by simulation result? The theorem says every minimax value function is an optimal value function, which is a mathematical fact as proved. Maybe we didn't get the reviewer's point here, in that case it would be helpful if the reviewer could further elaborate a bit about this question.

---

> ### Author Response · Authors · 2021-11-17
> **Regarding algorithms and numerical results**
>
> This response is dedicated to the reviewer's concern 4,5,6.
>
> (1)"*Besides algorithm1, is there a general algorithmic framework based on this nonlinear.*"
>
> Thanks for the question. The key idea of our algorithm is to approximate the minimax value functions via minimizing the Bellman-Lagrangian function under the stationary distribution of an optimal policy. Depending on how the gradient of the Lagrangian is specifically estimated (as well as on how the stationary distribution is sampled), one may make her own trade-offs to design different algorithms under this general idea. The specific algorithm shown in our submitted paper is just an example of such algorithm which demonstrates a "vanilla" version of the Lagrangian gradient estimator. In this sense, we indeed consider our algorithmic idea as a "general" algorithmic framework which is in analog to Policy Gradient in RL -- the latter similarly gives a blueprint of learning (including the objective function, the gradient-based optimization method, as well as a vanilla version of the policy-gradient estimator).
>
> We have uploaded a revised version of the paper, in which we substantially revised Section 4, as well as Appendix D, to more explicitly reflect the above perspective. The revised section 4 now starts with giving the "general idea" of Lagrangian minimization, which we named it as LAMIN in the revised version, which is followed by two "specific algorithm variants" under the LAMIN idea. The original algorithm in the submitted paper is called LAMIN2 in the revised paper.
>
> **Besides that, in the revised paper we've added another LAMIN algorithm, called LAMIN1**. LAMIN1 is closely related to the original algorithm 1 (a.k.a. LAMIN2 now), while giving slightly better performance in our experiment than the latter (see Figure 1 and Figure 2 in the revised paper for details). We hope this newly-added algorithm further facilitates the understanding/appreciation of Lagrangian minimization as a "general idea" that could potentially accommodates various specific algorithm designs.
>
> (2)"*There is no any convergence analysis of the proposed imitation learning algorithm.*"
>
> As mentioned above, the key idea of our algorithm is to figure out a good learning objective (i.e. the Bellman-Lagrangian function) to optimize. The optimization procedure itself is just a standard stochastic gradient descent process which thus shares generic and well-known convergence properties with all SGD algorithms. For example, the algorithm will converge to a local minimum with probability 1 if the learning rate satisfies $\sum_t \alpha_t = +\infty$ and $\sum_t \alpha_t^2 < +\infty$. We have added this remark in Appendix D.1 of the revised paper, in the paragraph right below Algorithm 1.
>
> (3)"*Numerical results are limited. The ELP example shown in Figure 2 is rather artificial.*"
>
> First, the ELP example in Figure 2 is not meant to be a numerical result (although we can see integers like 1,2,3 there...); instead, it serves as a counter-example to *logically* prove the suboptimality of minimax value functions.
>
> We consider our WMT experiments as the main numerical results of this paper. Comparing with many RL/IL works, our numerical result has the following strengths:
>
> -- The machine translation problem itself is a highly impactful downstream application. Breakthroughs in this area often lead to immediate and significant real-world impact (e.g. see "Google's Neural Machine Translation System, 2016").
>
> -- Moreover, machine translation is also a quite challenging task. In fact, MDP-based algorithms are not known to be effective in machine translation as of now, despite a very active research on this topic. To our best knowledge, **our WMT results as presented in Section 4 is one of the first empirical observation where a MDP-based solution can *independently* train a Transformer-scale neural network to reach/outperform SOTA performance in this problem**. We've added a paragraph in Section 6 (related works) in the revised paper to better explain this background.
>
> -- The WMT dataset we used is not a simulated environment; instead, the data is generated by human translators in real-world transactions (such as in UN conferences). The dataset is constructed for the annual WMT competition (see http://www.statmt.org/wmt21/), with the goal of powering large-scale training of SOTA neural networks in mind.
>
> Lastly, we kindly remind that this paper is mainly positioned as a "theory paper" where the main goal is to characterize a theoretical phenomenon (i.e. the Lagrangian duality phenomenon in episodic learning). The algorithmic/numerical part of the paper is mostly for demonstrating the empirical implications of our theory, which, we would argue, is a strength of this paper comparing with the many theory papers accepted by ICLR. With both theoretical and empirical results reported in the current paper (which leads to a 29-page manuscript), we plan to add more empirical results in our future works.

---

### Official Review · Reviewer_1YUz · 2021-11-08

**Correctness:** 3
**Technical Novelty And Significance:** 2
**Empirical Novelty And Significance:** 3
**Recommendation:** 5
**Confidence:** 3

**Main Review:**

1. One motivation for this work is to work with Q-functions, rather than V-functions; the latter readily admit LP reformulations as the paper acknowledges. Usually, the advantage of having Q-functions is that one can immediately construct an (implicit) greedy policy out of it. But such an advantage is not specific to Q-functions if one has a simulator (i.e. can take samples from s' ~ s,a). Since creating a simulator on the translation task is straight-forward (dynamics are explicitly known), why is then the translation task a suitable application to demonstrate the efficacy of this approach?
2. It is hard to make a case for the distinction in results for MDP with terminal states vs. the discounted setting. Do the results here specifically make use of this distinction? (Is Theorem 4 a part of this answer?) Theorem 1, for example, essentially follows from B^n (for large enough n) being contractive for MDP with terminal rewards, which I would argue is sufficiently similar to the disocunted case -- the same proof structure goes through. An interesting consideration here would be if only for select policies (vs. all policies currently) was termination guaranteed.
3. The policy optimization procedure of taking gradients while replacing max_a with some heuristic (like using the behavior policy) is a bit unsatisfactory. Why is such a substitution justified? In general, granted this formulation is non-linear, when is it solvable? From my understanding, this issue is also why Nachum, Dai 2020 take the gradient on the policy evaluation lagrangian (rather than for the optimal one).
4. Lastly, which, if any, of these results translate when the domain of Q is some restricted class of functions?

**Summary Of The Paper:**

This work work in the setting of MDP with terminal states, and considers a lagrangian relaxation (really a reformulation) on the Q-values. While the fact that LP on V-functions and the occupancy measure LP are equivalent (without function approximation) follows from LP duality, the equivalent Q-value formulation is non-linear; the paper nevertheless shows that for the lagrangian strong duality holds.

**Summary Of The Review:**

I'd be willing to raise my score if the above points are addressed. Otherwise, I feel the paper proves something interesting, but the translation of these ideas into either experiments or downstream theory is lacking.

---

> ### Author Response · Authors · 2021-11-10
> **Regarding reviewer's point 1 (Q-based approach vs V-based approach)**
>
> This response is dedicated to the reviewer's concern on why the translation task is suitable to demonstrate the efficacy of the Q-based approach given that the transition dynamic is known in this task which enables the V-based approach too.
>
> To answer this question, let us first make distinction between two statements:
>
> Statement 1: Q-based algorithms which does not require transition model (thus are model-free) are generally more favorable than V-based algorithms which require transition model (thus are model-based) in many applications.
>
> Statement 2: Our specific Algorithm 1 is a competitive model-free algorithm on large-scale AI benchmark.
>
> (1) **In our current paper, the translation experiment is designed to support statement 2, not statement 1**, and we think our experiment indeed well supported this statement for the following reasons:
>
> -- The WMT14 en2de dataset is a highly influential testing benchmark that has proven to be discriminative in validating promising solutions. For example, the classic "Attention Is All You Need" paper demonstrated the efficacy of the Transformer model by showing 2 BLEU score improvement on this benchmark; the model later becomes pervasive. In our experiment we showed 1 BLEU score improvement over that paper.
>
> -- Translation itself is a task of immediate real-world usage, compared with many game-based (or simulator-based) RL/IL benchmarks. The data in our benchmark are gathered from translation records in real-world transactions.
>
> -- The benchmark we used is highly challenging. As far as we know, no MDP-based algorithm (even model-based) has demonstrated competitive learning performance on this benchmark as our Algorithm 1 does. In particular, we are not aware of comparable results for V-form Lagrangian based algorithms.
>
> (2) Is it suitable to use an experiment that does not best support statement 1 for validating statement 2 (only)? We think so -- the WMT benchmark is a valid testing bench for model-free algorithm even though the transition model is actually known (thus not a best demonstration of the advantage of Q-based algorithms). This is in analogy to many RL benchmarks, such as Atari games and Mujoco-based locomotion simulation, where the transition dynamic is also known (because the simulator code is available, and some research indeed utilize the simulator to study model-based algorithms, such as MCTS-based ones). Not too many people worry about the validity of those RL benchmarks for testing model-free algorithms, even though they do not best demonstrate the power of model-free algorithms (as you can still run model-based algorithm in those benchmarks).
>
> (3) Now, if the translation experiment supports only statement 2, how is statement 1 supported? Our answer is that we think statement 1 is more or less a consensus in the community that does not necessarily require special support in our paper. While we agree that, as the reviewer mentioned, V-function can be used for decision making *if* the transition dynamic can be simulated, to get ride of the assumption/condition of having the transition model is all model-free learning is about. This is why we only see Q-Learning algorithms, not V-Learning algorithms, in the RL textbooks (while V-based value iteration is popular in DP). In general, the environment dynamics can often be elusive to capture in the real world, which justifies the study of model-free algorithms in general, and of Q-based approach in our case specifically. Even for translation, a real-world variant of it is *simultaneous translation* in which the source sentence is not entirely given at the beginning but is gradually completed while the translation goes on, which naturally creates an unknown transition dynamics. Q-based approach keeps all the flexibility and scalability to learn in an unknown world, so is generally favorable (when we can figure out how to make it works).
>
> (4) Despite the above, **we would nonetheless like to point out a practical barrier in applying the (theoretically applicable) V-based approach to the exact translation problem as considered in our paper**. As the reviewer mentioned, since we can easily generate $s'$ from $(s,a)$ in translation, we can simply compare $V(s')$ for the successor state $s'$ of all $a$, which is certainly doable. However, how long would this take in practice? There are $37,000$ possible actions in our problem, so the V-based approach would need to run a fairly large deep neural network on $37,000$ different $s'$ for generating just one token. In comparison, the Q-based approach -- such as our implementation of Algorithm 1 -- enjoys the highly efficient NN architecture inherited since DQN time which takes only a single state $s$ as input and a single $37,000$-dimension vector as output to obtain all the Qs. Thus, although the V-based approach is **computable** in theory, it would at least need some new design on the NN architecture to make the computation **efficient** in practice.

---

> ### Author Response · Authors · 2021-11-11
> **Regarding reviewer's point 2 (episodic setting vs discounted setting)**
>
> This response is dedicated to the reviewer's concern on motivation to work on episodic learning setting, instead of discounted-MDP setting, the "industry standard", especially given that our technical results/ideas seem not unique/distinct to the former but might possibly hold in the latter too.
>
> (1) We've explained in the paper our motivation of working on episodic setting but let us elaborate more. We indeed consider our shift to the episodic setting as a merit of the paper, but we don't shift for the purpose of just shifting to a new setting. The justification is *not* that we cover yet another new MDP variant that's not explored by previous works before (and thus broaden the scope of applicability for the Lagrangian method) -- no, that's not our point (and had that been the case, we'd agree that showing something distinctly special in that new MDP variant would have been important).
>
> Instead, we study the episodic setting because it is really the scenario in so many real-world practices. The episodic setting deserves focused study not because it's "theoretically distinct", but because it's empirically highly relevant. While the discounted setting does capture certain economic/psychological applications, much more current AI practices deal with episodic learning tasks instead (e.g. atari games, locomotion, route navigation, Go/chess, translation, dialogue system, image captioning, etc). The translation problem considered in our paper is an example, where the termination is indeed guaranteed and the discounted objective is indeed inaccurate (if not actually go the opposite way with the spirit of the BLEU metric, the true objective function used in machine translation) -- see our Appendix D.2, as well as footnote 2 in page 7.
>
> We note that we are not the only advocates to shifting away from the discounted setting. For example, (Sutton&Barto, 2018) dedicated its Section 10.4 on this issue, and the DP community has subsumed the discounted setting as a special case of an finite-termination setting for long time (e.g. see "Neuro-Dynamic Programming" by Bertsekas). Our paper contributes to this movement, closing a gap between theory and practice.
>
> (2) We stress that our main results are not known (even) in the discounted setting, so it's really that the results only *possibly hold* in discounted setting, but not *already known to hold* there. To be precise, by "main results" we mean
>
> i) the strong duality property for minimax-type saddle points of the nonlinear Q-form Lagrangian (Lemma 2.2, Theorem 2, Proposition 3)
>
> ii) the Lagrangian-based imitation learning algorithm and its competitive performance in machine translation (Algorithm 1, Table 1, Figure 1)
>
> iii) the sub-optimality of minimax Lagrangian saddle points, the introduction of maximin points and the proof of their optimality (Figure 2, Lemma 4.1, Theorem 4)
>
> To our best knowledge, result i), ii), iii) would be novel even in the discounted setting (had we developed the results there).
>
> (3) The ideas/tricks we used to derive these results are also mostly new, and we indeed utilized special structures of the episodic setting from time to time:
>
> -- For result i), strong duality for the linear V-form Lagrangian, or for the linear Q-form policy-evaluation Lagrangian, directly follows from the generic LP duality. In contrast, the strong duality we demonstrated relies on the special form of the Lagrangian (Lemma 2.2), which in turn replies on the idea of averaging over terminal states/actions (instead of over initial states as before), and on the conception of "conjugate policy" as entailed by the usage of terminal states. This general logic flow does not assimilate with anything we are aware in related works. Our proof utilized an *ELP ergodic theorem* which is unique to episodic setting.
>
> -- For result ii), we are not aware of any algorithm derived from unilaterally minimizing the Lagrangian as our Algorithm 1. Plus, the successful application of Lagrangian-based algorithm (or MDP-based algorithm in general) to machine translation should be something new too (so far RL/MDP only plays marginal role in MT).
>
> -- For result iii), it appears that the sub-optimality of minimax saddle points has been overlooked in previous works (e.g. see [Cho and Wang, 2017], [Dai et al, 2018], [Nachum and Dai, 2020]), so our treatment on maximin points should be completely new. In particular, the symmetry breaking between minimax and maximin points concluded from this treatment may challenge a tradition in the area since 1980s.
>
> In summary, the motivation of working on episodic setting is not for technical convenience but for its real-world relevance, and our main results/ideas are novel regardless of the select MDP setting. Some part of our work does rely on unique properties of ELP but we don't claim that similar results cannot hold in discounted setting -- in any case, such potential generalizability should be a merit of a work, instead of a drawback.

---

> > ### Author Response · Authors · 2021-11-11
> > **continued response to reviewer's point 2 (on Theorem 1)**
> >
> > In point 2, the reviewer took our proof of Theorem 1 as an example, arguing that our proof structure assimilates with the discounted case in terms of the idea of proving the contraction property, and suggested the cases where only some policies are proper.
> >
> > Firstly, we want to clarify that we didn't claim our proof of Theorem 1 as a highlighted contribution of the paper, which is why this 6-page-long proof is entirely placed in appendix. Actually we did *not* even consider the conclusion of Theorem 1 itself a main result of the paper (although the theorem is indeed a new result, to our best knowledge). Instead, we consider the fact that "*all the main results of the paper is built on top of the episodic setting which is enabled by the conclusion of Theorem 1*" as a major merit of the paper (for reasons we've elaborated above). So, as a *supportive result*, the proof structure of Theorem 1, regardless of its similarity to traditional ones, may not matter too much regarding whether our *main result* makes special use of the episodic setting or not (i.e. reviewer's point 2), where "main result" refers to the result i), ii), iii) in our response above.
> >
> > On the other hand, and as a matter of fact, our proof of Theorem 1 does differ from the traditional proof in discounted setting in the following two aspects:
> >
> > -- Although both reduce the fixed-point property to the contraction property as the reviewer pointed out, the reason behind the contraction is totally different (and to prove the contraction makes up bulk of the overall proof). In discounted-MDP, the constant discounting factor $\gamma$ immediately guarantees contraction, and Eq.(14) is pretty much all what we need in that case (the whole proof would be just several lines long). In episodic setting, however, the contraction property is rooted from the *graph property* as identified in Lemma 1.1, and also the overall logic to rigorously prove the contraction using this graph property is substantially more complex.
> >
> > -- In discounted setting, it is the Bellman operator $B$ that is proved contractive, which immediately entails the fixed-point property of $B$ itself. In contrast, in episodic setting, it is the composited operator $B^n$ that is proved contractive, but what we want is to prove about the operator $B$. Additional logic is needed to fill this gap (see Lemma 1.3).
> >
> > To our understanding, the technical simplicity to prove Theorem 1 in the discounted setting was historically a main reason that explains the popularity of discounted-MDP. Proving equivalent theorem in undiscounted settings is known to be notoriously harder (e.g. see [Nachum and Dai, 2020], section 7).
> >
> > "*An interesting consideration here would be if only for select policies (vs. all policies currently) was termination guaranteed*"
> >
> > Thanks for the suggestion! As far as we know, this is indeed an active topic in the DP community, where only a subset of policies are assumed to be proper (i.e. finite terminated). However, as we explained above, the motivation of the episodic learning setting is mainly for its real-world relevance. From learning perspective, it seems that having all policies finitely terminated is not just a mathematical assumption but is essential for effective learning at all -- otherwise the learning data may catastrophically degenerate when it runs into an infinite-long episode (which is not a problem in the DP context, though, thus justifies the partial-termination setting there).

---

> ### Author Response · Authors · 2021-11-11
> **Regarding reviewer's point 4 (function approximation)**
>
> In this response we answer the reviewer's question on applicability of our results when Q-function is represented by a function approximator such as deep neural network.
>
> "*which, if any, of these results translate when the domain of Q is some restricted class of functions?*"
>
> First, the entire Section 4 is assuming approximated Q-function already. Our algorithm is proposed in the context of parameterized Q-functions *by design*, and is tested on standard deep learning benchmark. Our experiment used state-of-the-art deep neural network (the Transformer model), which consists of ~25 million parameters.
>
> For Section 3, Lemma 2.2 gives a useful dual form of the Lagrangian in a per-Q sense, thus this result remains intact under any restricted class of Q-functions. The duality result observed in this section directly led to the deep learning algorithm in our experiment, which is a demonstration of our theory's potential to inspire practical solutions even when Q is restricted to highly complex sub-classes.
>
> The theory developed in Section 5 concerns issues at a more basic level, discussing what value function *should* be the target of learning, which include learning with parameterized Q-functions. In particular, the sub-optimality of minimax Lagrangian saddle points as shown in this section is a limit of this approach even when the domain of Q is the entire value space, thus is also a limit when the Q-domain is a restricted class. Our observations about the optimality of maximin value functions, and about the symmetry breaking in learning settings for the primal-form vs dual-form Lagrangian method, point to a new direction to design learning algorithms for parameterized Q-functions (for several tens of years people have been exclusively looking at the primal-form optimization).
>
> Admittedly, it's possible that these minimax-type or maximin-type saddle points, or even the Bellman optimal value $Q^*$, could all fall outside the restricted Q-domain at all, or are not necessarily attained when the underlying stochastic-gradient optimization fails to find the global optimum, in which case our current paper didn't shed too much light on how to approximate these saddle points or predict how well such an approximation would perform. But this is arguably a rather general issue that most value-based methods in the current literature would suffer at the moment, and is considered beyond the scope of this paper.

---

> ### Author Response · Authors · 2021-11-17
> **Regarding reviewer's point 3 (nonlinear optimization)**
>
> This response is dedicated to the reviewer's concern on the challenges of nonlinear optimization in our proposed algorithmic idea.
>
> (1)"*The policy optimization procedure of taking gradients while replacing max_a with some heuristic (like using the behavior policy) is a bit unsatisfactory. Why is such a substitution justified?*"
>
> The substitution is formally justified by Proposition 4 in Appendix D of our submitted paper; the proposition is moved to the main text of Section 4 in the revised paper, given the reviewer's concern. In the algorithm (which is now called LAMIN2 in the revised paper), max_a is not replaced with just *a* behavior policy; instead, the behavior policy is a special one that guarantees consistency/unbiasedness of the gradient estimator "under mild assumption" in theory, and "for most of the time" in practice. Specifically, Proposition 4 proves that the substitution is precisely unbiased (in a per-step sense) as long as the best action as suggested by the Q-function is unique -- even a gap of $10^{-10}$ in the values of the best and second best action would satisfy this assumption. This is arguably a fairly reasonable assumption for continuously-valued and sophisticatedly-parameterized Q-functions, especially if the function parameter is stochastically generated, say, by a SGD procedure.
>
> Moreover, we kindly note that in the revised paper (Section 4), we've discussed another gradient estimator, called LAMIN1 (the original algorithm that used the substitution is called LAMIN2 now). LAMIN1 directly takes the gradient of a smoothed Lagrangian function that is perfectly differentiable. The smoothed Lagrangian subsumes the original Lagrangian as a limiting case, and can be arbitrarily close to the latter under a small enough temperature $\beta$. More importantly, applying the $\beta$-smoothing in the optimization objective as LAMIN1 does (instead of in gradient estimator) may actually correct the suboptimality bias of minimax value functions; see Appendix D.1 in the revised paper for details. Empirically, the new algorithm (LAMIN1) slightly but consistently outperforms the old algorithm (LAMIN2) on the WMT benchmark (see Fig 1 and 2 in the revised paper for details).
>
> (2)"*In general, granted this formulation is non-linear, when is it solvable?*"
>
> Thanks for the question. In general, we don't feel that the nonlinear optimization objectives in our treatment are particularly troublesome, not especially when deep neural networks are used, in which case the optimization is *always* nonlinear anyway, no matter if it's V-based or Q-based, policy-evaluation or policy-optimization. The advance of deep learning has given us pretty practical techniques to deal with such nonlinear functions.
>
> The issue around max_a is actually an example supporting our point above: the problem of differentiating something like max_a is *not* unique to our algorithm; instead, the very popular ReLU unit in deep neural networks already brought in such max operator in the objective function of every deep learning algorithm (that uses ReLU networks). The way our LAMIN2 algorithm deals with this issue is essentially based on the same idea as how people deal with ReLU -- the max operator only makes the objective function non-differentiable at some exceptional points whose impact is empirically limited.
>
> In our view, the main challenge is not at the optimization side, but to derive a proper objective function from the nonlinear formulation which, if ever possible, would usually crucially rely on some special mathematical structures -- such as the Theorem 2, Theorem 4 (=Theorem 5 after the revision) and Lemma 2.2 in our paper -- and revealing the existence of such structures for the Lagrangian function of episodic learning problems is precisely the main goal of this paper. As an example, our Lagrangian minimization algorithm is a direct consequence of the strong duality as proved by Theorem 2 -- without the minimax theorem, there is no logical basis to minimize Q under a fixed $\lambda$ (which is the basic idea of our algorithm). Conceptually, a main message our paper intends to deliver is that, **the nonlinear optimal-Q Lagrangian is not just "a nonlinear formulation" but it enjoys some special structures which make the formulation not as intractable as we might think previously**.
>
>
> (3)"*this issue [that there is a max_a] is also why Nachum, Dai 2020 take the gradient on the policy evaluation lagrangian (rather than for the optimal one).*"
>
> We are not sure that adapting their theory to the optimal-Q setting would only have issue in optimization; afterall, it's not clear that the duality argument in [Nachum, Dai 2020] still applies to the optimal-Q Lagrangian which is neither linear nor convex. On the other hand, it could be an interesting future work to combine our optimization tricks for max_a with their primal-dual framework, if differentiating max_a is a main issue there.

---

### Author Response · Authors · 2021-11-17
**The paper is revised**

Dear Reviewers,

Thank you for the comments on our paper. We have posted detailed responses to each and every issue raised by you. The paper is also revised, with the list of changes summarized below:

-- Section 4 (algorithms & experiments) is substantially revised and restructured, which now makes a more clear separation between our ideas about "objective function design" and our ideas about "optimization & differentiation". Appendix D is also revised accordingly.

-- In section 4, a new algorithm (called LAMIN1) is added, which is based on the same algorithmic idea of Lagrangian minimization, but differ from the existing algorithm (called LAMIN2) in how the gradient of Lagrangian is estimated. Appendix D.1 gives extended discussion on the more principled rationale of LAMIN1, as well as on its convergence property.

-- In section 4, similar experimental analysis are conducted and reported for the new algorithm LAMIN1, which showed slightly better performance than the existing algorithm LAMIN2, increasing the margin of our MT result against the baseline to 1.4 BLEU score now.

-- Proposition 4, the theoretical justification of our existing algorithm LAMIN2, is moved from appendix D to the main text of section 4.

-- Section 6 (related works) is extended, so as to better explain the significance of our results (theoretical + empirical).

-- In Page 2, we added a footnote to emphasize the equivalence of the state-based reward formulation as used in our paper with action-based reward formulations.

-- In appendix B.3, we added some more elaboration on the complementary slackness argument in the proof of Theorem 2.

We hope our responses and the revised paper have completely resolved all of the concerns mentioned in the reviews, and hope they help you to better understand and appreciate this paper. We are happy to answer more questions in the rest of the discussion period.

Thanks!

---

### Public Comment · ~Huang_Bojun1 · 2022-07-19
**"Discussion period" no discussion, "open review" not open**

As the author, I believe this paper has received low-quality reviews in ICLR 2022. Specifically, I made significant efforts to address the reviewers' questions/concerns in **11 rebuttal posts**:

-- In the rebuttal posts, I provided extensive answers to questions from Reviewer 1 who wrote "*I'd be willing to raise my score if the above points are addressed*".

-- In the rebuttal posts, I gave pointers to specific paragraphs in the paper which explicitly discussed the topics that reviewer 2 claimed are missing (and deemed as the main issue of the paper).

-- In the rebuttal posts, I pointed out factual mistakes in reviewer 3's (mis)understanding on an issue of our paper that the reviewer believed is "*the major weakness of this paper ... [that] tremendously restricts the usefulness of this paper*".

But I received **zero reply** in the (so-called) "discussion period" of the review process. I wrote my concern about this situation to the meta-reviewer around the end of the discussion period, but the meta-reviewer was not responsive too. It seems to me that the meta-reviewer did nothing throughout the review process but leaving a very ambiguous meta-review saying that "many things are not clear, and please re-submit" -- a paper is never clear unless you read it!

The paper is accepted to ICML 2022. See https://proceedings.mlr.press/v162/bojun22a.html

---

### Decision · Program_Chairs · 2022-01-20

**Decision:**

Reject

**Comment:**

Although the submission studies an interesting question, many parts are not clear enough and the presentation needs to be improved. I encourage authors to revise the paper accordingly and resubmit in future venues.